# Single-molecule visualization of conformational changes and substrate transport in the vitamin $B_{12}$ ABC importer BtuCD-F

Joris M.H. Goudsmits[1], Dirk Jan Slotboom [1,2,3] & Antoine M. van Oijen [1,2,3,4]

ATP-binding cassette (ABC) transporters form the largest class of active membrane transport proteins. Binding and hydrolysis of ATP by their highly conserved nucleotide-binding domains drive conformational changes of the complex that mediate transport of substrate across the membrane. The vitamin $B_{12}$ importer BtuCD-F in *Escherichia coli* is an extensively studied model system. The periplasmic soluble binding protein BtuF binds the ligand; the transmembrane and ATPase domains BtuCD mediate translocation. Here we report the direct observation at the single-molecule level of ATP, vitamin $B_{12}$ and BtuF-induced events in the transporter complex embedded in liposomes. Single-molecule fluorescence imaging techniques reveal that membrane-embedded BtuCD forms a stable complex with BtuF, regardless of the presence of ATP and vitamin $B_{12}$. We observe that a vitamin $B_{12}$ molecule remains bound to the complex for tens of seconds, during which several ATP hydrolysis cycles can take place, before it is being transported across the membrane.

[1] Zernike Institute for Advanced Materials, University of Groningen, Nijenborgh 4, 9747 AG Groningen, The Netherlands. [2] Groningen Biomolecular Science and Biotechnology Institute, University of Groningen, Nijenborgh 4, 9747 AG Groningen, The Netherlands. [3] Centre for Synthetic Biology, University of Groningen, Nijenborgh 4, 9747 AG Groningen, The Netherlands. [4] Present address: School of Chemistry, University of Wollongong, Wollongong, NSW 2522, Australia. Correspondence and requests for materials should be addressed to D.J.S. (email: d.j.slotboom@rug.nl) or to A.M.v.O. (email: vanoijen@uow.edu.au)

ABC transporters are membrane proteins that translocate substrates across a lipid bilayer[1]. They consist of two highly conserved nucleotide-binding domains (NBDs) that utilise energy of ATP binding and hydrolysis to drive conformational changes in the two transmembrane domains (TMDs), resulting in the formation of a pathway for substrate transport[2]. The TMDs are not conserved among all ABC transporters, and appear to have evolved from multiple ancestors[3]. Based on the different three-dimensional architectures of the TMDs, ABC importers can be categorised as type I, type II or energy-coupling factor (ECF) transporters[4–6], each with a different mechanism of transport[7–9]. Type I and II importers are complemented with soluble substrate-binding proteins (SBPs) that bind their ligands and deliver them to the TMDs. Based on structural properties, these SBPs can be further classified into clusters[10,11].

The *Escherichia coli* (E. coli) vitamin B$_{12}$ transporter BtuCD-F is the best characterised type II importer. The homodimer BtuC spans the membrane and the two identical cytosolic ATPase domains BtuD form a sandwich dimer that couple chemical

energy of two ATP molecules into structural changes of the full complex[12]. A single substrate-binding protein BtuF completes the transporter. This SBP belongs to cluster A or class III and exhibits relatively small conformational changes upon substrate binding[13]. Extensive structural and biochemical characterisation of BtuCD alone or in complex with BtuF has provided the framework for understanding the mechanism of vitamin B$_{12}$ transport: crystal structures have revealed several intermediate states in the transport cycle[14–16], the gating mechanism of the TMDs upon ATP hydrolysis has been investigated with EPR techniques[17,18], and biochemical characterisation of the complex in detergent and proteoliposomes has given insight into the molecular steps underlying transport[19,20]. These studies all suggest that BtuCD-F employs a different mechanism than that described by the alternating-access model for type I importers. Nonetheless, many questions remain unanswered with different studies in detergent and lipid environments showing varying results. Does the soluble substrate-binding protein BtuF remain bound to the BtuCD transmembrane complex during the cycle[19], or does ATP hydrolysis release the binding protein into the periplasmic

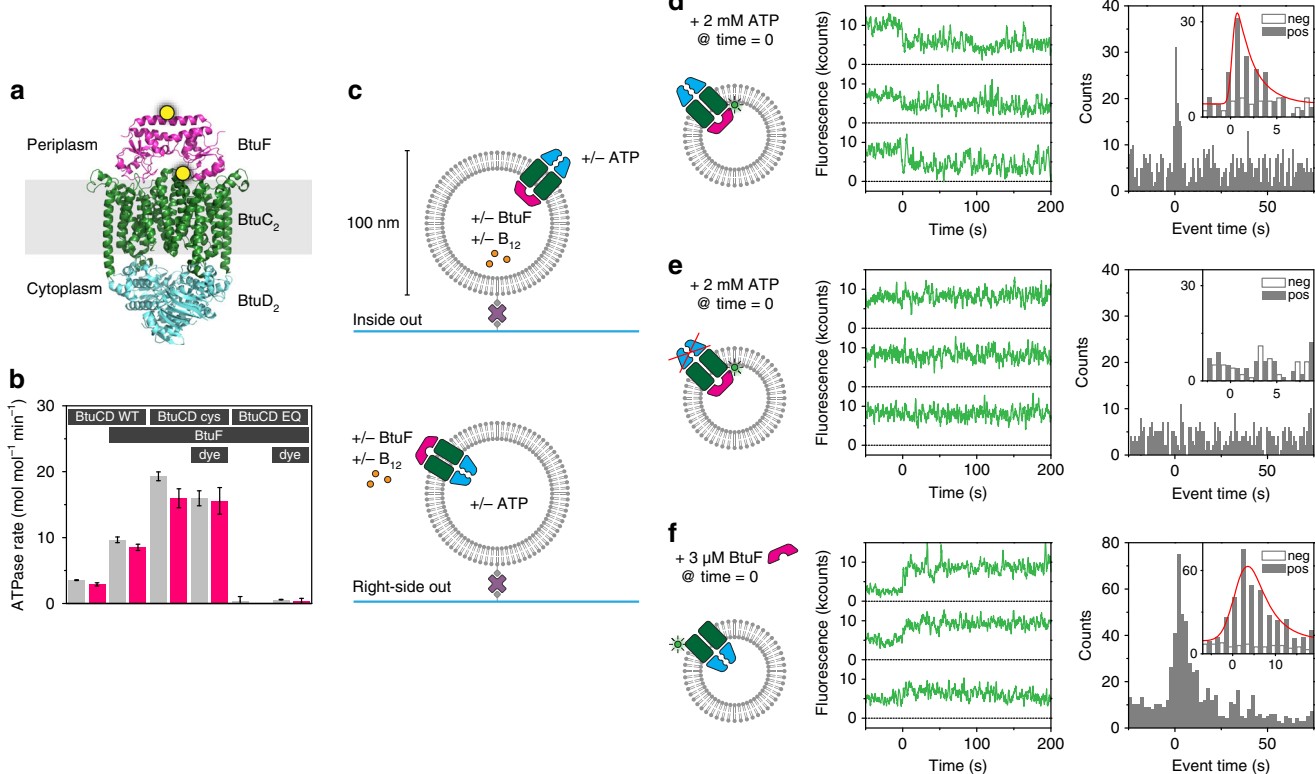

**Fig. 1** Experimental setup, ATPase rate and fluorescence quenching by ATP and BtuF. **a** Structure of BtuF (pink) bound to BtuCD (BtuC homodimer: green, BtuD homodimer: blue) in the absence of nucleotide and substrate (Protein Data Bank (PDB) ID 2QI9). Cysteine mutations for labelling in BtuF (D141C) and BtuC (Q111C) are marked with a yellow dot. The distance between the labelling positions is ~37 Å. **b** ATPase rate of various BtuCD-F mutants with or without fluorescent label reconstituted in liposomes and loaded with (pink bar) or without (grey bar) vitamin B$_{12}$. BtuCD WT denotes the wild-type, BtuCD$_{cys}$ denotes the cysteine mutant and BtuCD$_{EQ}$ denotes the cysteine mutant that is ATPase-impaired. For all combinations, the cysteine mutant of BtuF is used. Measured rates are not corrected for orientation of the transporter. When BtuF is present at the concentrations used, the full complex is formed. Values displayed are the mean and standard deviation of three experiments. **c** Experimental design (fluorescent labels are omitted). BtuCD was reconstituted in liposomes of 100-nm diameter in ratios such that on average one transporter was found per liposome. By introducing BtuF and vitamin B$_{12}$ to the lumen of the vesicle and ATP on the outside or vice versa, only one particular orientation of the transporter was probed. Proteoliposomes are tethered to a glass surface via a biotin-streptavidin link and imaged using TIRF microscopy. **d** A complex of BtuCD$_{cys}$ labelled with Alexa Fluor 555 and unlabelled BtuF showed decrease in fluorescence intensity upon addition of 2 mM ATP and 10 mM Mg$^{2+}$ on the outside (middle panel). The distribution of event times of the first drop of intensity is plotted in a histogram (right panel) for the positive (pos, with ATP) and negative (neg, without ATP) experiment. For a description of the data analysis, see methods. **e** Similar experiment as described in **c**, but with the ATPase impaired mutant BtuCD$_{EQ}$. No events were observed. **f** Similar experiment as described in **c**, but the vesicle lumen was left empty. Upon introduction of BtuF to the outside of the liposomes an increase in fluorescence intensity was observed. For each condition in **d**–**f** around 1000 single-molecule fluorescence traces were analysed

space[20]? Is the substrate immediately transported, and is ATP merely required to reset the transporter[20]? Why is the ATPase activity at least one order of magnitude higher compared to transport of vitamin $B_{12}$—in other words, why is there no strong coupling between ATP hydrolysis and substrate translocation such as observed for the well-studied type I maltose importer MalFGK[21]? In order to address these questions, we have employed single-molecule fluorescence techniques to follow individual BtuCD-F proteins reconstituted in liposomes through time and to directly observe steps of the transport cycle. Ultimately, we interpret our results in the light of different existing models for transport by BtuCD-F.

## Results

**Single-molecule experimental setup**. To visualise BtuCD and BtuF, we created single cysteine mutants of BtuC (Q111C, on the periplasmic loop connecting transmembrane (TM) helix 3 and 4) and of BtuF (D141C, pointing outward in the middle of the alpha helix connecting the two lobes) that allow for specific coupling of fluorescent labels to each of these proteins (Fig. 1a). We also introduced a tryptophan mutation in BtuC (W115L) to remove any quenching effects of this tryptophan on our fluorescent probes[22]. The labelling positions allow for different observation strategies that will be individually explained in detail below. ATPase and radiolabelled uptake experiments demonstrated that mutations and labels had no critical effect on ATP hydrolysis and transport activity (Fig. 1b, Supplementary Fig. 1). In the background of the cysteine mutant of BtuCD (termed BtuCD_cys), we also created an ATPase impaired mutant by an additional mutation in the walker B loop (E159Q) of BtuD (termed BtuCD_EQ) (Fig. 1b). Based on our previous work[23], we designed a procedure for reconstituting and visualising single BtuCD proteins in liposomes with a diameter of 100 nm (see Methods). Using a fluorescence quenching method in bulk, we determined that 55% of the transporters were orientated right-side out and 45% inside out (Supplementary Fig. 2). Although we do not have an assay to directly examine the orientation of each single BtuCD protein reconstituted in a single liposome, we were able to selectively analyse transporters that are in either orientation in the membrane, because components, such as ATP, vitamin $B_{12}$ or BtuF, that are needed for transport, each exclusively interact with the cytoplasmic or periplasmic side of the transporter[15,19]. By introducing these components either in the lumen of the liposomes or on the outside, we were able to selectively probe with single-molecule sensitivity only those transporters that are placed in either the right-side-out or the inside-out orientation in the membrane (Fig. 1c). Proteoliposomes immobilised on the glass surface (see Methods) of a flow channel (Supplementary Fig. 12) were imaged using a home-built total internal reflection fluorescence (TIRF) microscope.

**Fluorescence changes on BtuCD induced by BtuF and ATP**. We set out to probe the interaction between BtuCD and BtuF in the absence of substrate and the dependence of this interaction on the presence of saturating concentrations of ATP. In detergent solution the proteins form a stably associated complex[20], but the stability of the complex in membranes is not known. For this set of experiments, we used quenching of a single dye as our observable for protein conformation. We labelled BtuCD with Alexa Fluor 555 and reconstituted it in such a way that the majority of liposomes contain at most one copy of the complex (Supplementary Fig. 3). The lumen of the proteoliposomes was loaded with an average of one molecule of unlabelled BtuF (equivalent to a concentration of 3.2 μM in a liposome with 100-nm diameter) (Supplementary Fig. 4). While the fluorescence was

high in the absence of nucleotide, in a fraction of liposomes (8% of the total) it decreased rapidly upon addition of ATP to the solution surrounding the liposomes. Taking into account that 45% of the transporters are oriented inside out (Supplementary Fig. 2), i.e. with the BtuD ATPase domains facing to the outside, only 18% of the complexes respond to ATP—or are active. This fraction of apparent active transporters corresponds well with the previously measured functionally competent fraction in bulk measurements[19]. Fluorescence traces of other (presumably inactive) complexes do not show any ATP-dependent features that can be discriminated from control experiments, and therefore we consider only the quenching as indicator for ATPase activity. The quenching of the fluorescence suggests an alteration of local protein environment of the dye and thus a conformational change of the complex (Fig. 1d, middle)[24]. Other dyes such as Cy3 and tetramethylrhodamine (TMR) manifest the same quenching behaviour. The distribution of quenching events plotted in a histogram (Fig. 1d, right) revealed an ATP response time in our flow cell of $1.9 \pm 0.5$ s (see Methods). When we performed the same experiment in the absence of BtuF, no events were observed (Supplementary Fig. 5), indicating that the observed decrease in intensity was mediated by interaction with BtuF. The same experiment with the ATPase impaired mutant BtuCD_EQ did not result in a change in fluorescence intensity upon addition of ATP (Fig. 1e). Similarly, the use of the slowly hydrolysable ATP analogue AMP-PNP instead of ATP[15], also showed no change in intensity (Supplementary Fig. 6). From these data, we conclude that ATP binding and hydrolysis alters the interaction between BtuF and BtuCD, changing the local environment of the dye and yielding a decrease in fluorescence. The experiment resulting in the starting state of the above experiment was also performed: unlabelled BtuF was added to the outside of proteoliposomes not containing ATP. In 45% of the cases (corresponding to 82% of the complexes with right-side out orientation), an increase in fluorescence intensity was observed, corresponding with the binding of BtuF to those BtuCD complexes that were oriented right-side out (Fig. 1c, bottom panel, and Fig. 1f). From these experiments, we can conclude that a strong interaction between BtuCD and BtuF resulted in a high fluorescence intensity, whereas a change in interaction induced by ATP hydrolysis lowered the fluorescence. These findings are well supported by ensemble measurements (Supplementary Fig. 7). Below we will use these intensity changes to probe ATP hydrolysis.

**ATPase activity by the BtuCD-F complex**. Based on the experiments presented in Fig. 1, we could not infer whether the change in fluorescence upon ATP hydrolysis was caused by complete dissociation of BtuF from BtuCD, or by more subtle conformational changes in the intact complex. In order to address this question, we used single-molecule FRET to probe the distance between BtuCD and BtuF. BtuCD_cys was labelled with a donor fluorophore (Alexa Fluor 555) and BtuF coupled to an acceptor dye (Alexa Fluor 647). Labelled BtuF was incorporated in the lumen of the liposomes. Based on the positions of the labels, as indicated by Fig. 1a, we expected a high FRET efficiency in the associated complex, but no FRET in case BtuF had dissociated from BtuCD. In the absence of ATP, high and steady FRET was observed (Fig. 2a, Supplementary Fig. 8), confirming that BtuCD-F is indeed a stable complex in membranes. By adding ATP in the absence of vitamin $B_{12}$, the total intensity decreased (Fig. 2b, middle panel), indicating quenching similar to what we observed with donor-only labelled protein (above). The observation that the measured acceptor intensity did not decrease to zero suggests that either cognate BtuF was not fully dissociated upon ATP hydrolysis, or that the binding protein

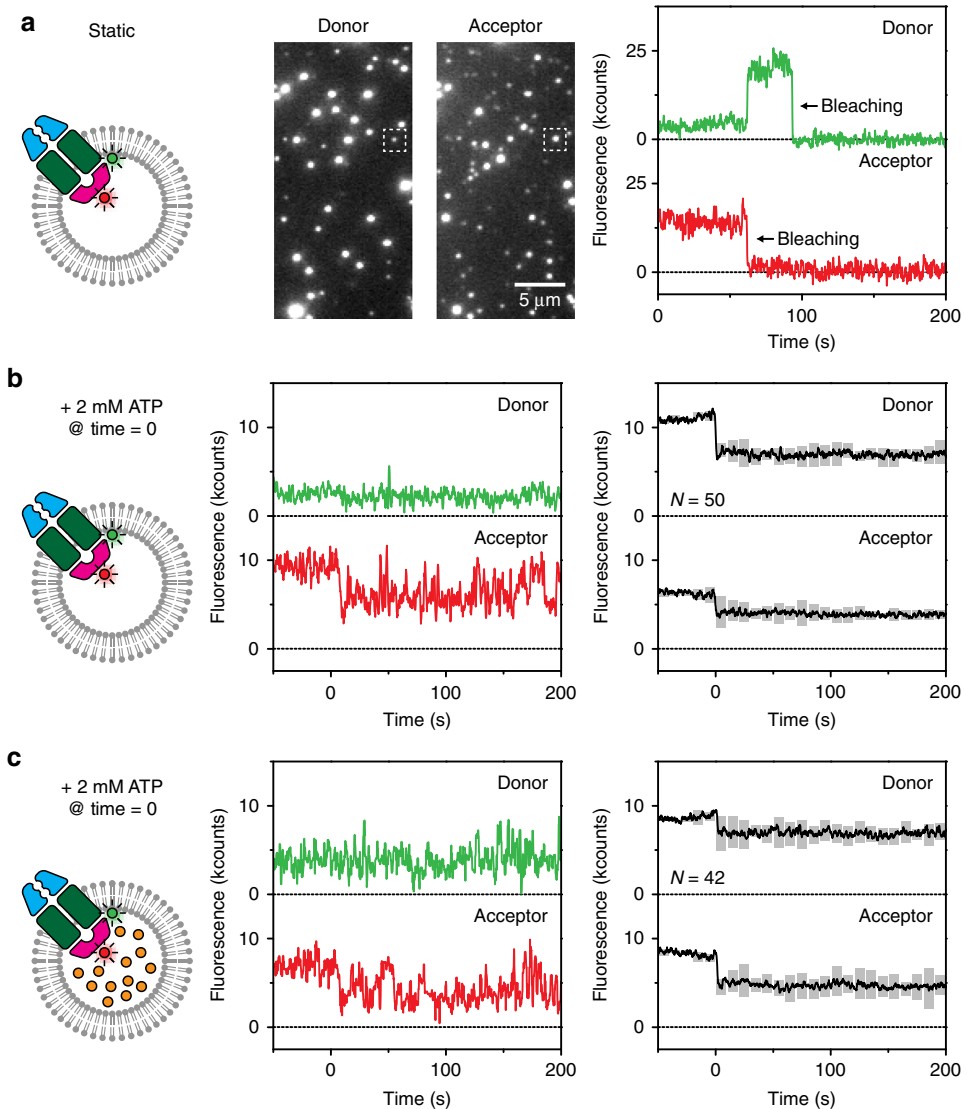

**Fig. 2** Complex formation observed with FRET. **a** A stable complex of BtuCD$_{cys}$ (Alexa Fluor 555) and BtuF (Alexa Fluor 647) was formed when reconstituted in substrate-free liposomes. The middle panel shows a section of a field of view. On excitation of donor fluorophores (left channel), emission of acceptor fluorophores was visible (right channel), which is an indication of FRET and thus complex formation. The right panel shows the fluorescence traces of the spots marked with a square. A 2.5× higher laser power density was used to promote bleaching of the dyes, here visible at ~65 and ~100 s. **b** The same complex as in **a** was used, but now ATP and Mg$^{2+}$ were introduced at time zero. The total intensity (sum of donor and acceptor fluorescence) decreased when ATP was present, and dynamics in the signal are visible (middle panel). The right panel shows the average of all traces where a drop in total intensity was observed upon introduction of ATP; the pair of traces shown in the middle panel is one of them. The grey floating bars indicate the number of times the signal exceeded a threshold (see Methods), and thus report on the extent of the fluctuations increasing at positive times. In total, over 1000 fluorescence traces were analysed. **c** Similar experiment as described in **b**, but with 100 μM vitamin B$_{12}$ introduced to the lumen of the vesicles

cycled between an associated and fully dissociated state on a timescale faster than our acquisition time. The observation of rare examples of exchange of labelled BtuF from the complex for unlabelled BtuF (and vice versa) on a timescale of minutes (Supplementary Fig. 9) indicates that the former case is more likely, as further supported in later experiments (described below). We also observed that the fluorescence intensity of mainly acceptor fluorophore fluctuated much more after the addition of ATP than before—from the lower fluorescence level almost back to the initial level (Fig. 2b, middle and right panel). The dynamics of the fluctuations in our traces did not depend on laser intensity and our observed fluctuations are on a timescale that is much larger than what can be expected for blinking (milliseconds). Therefore we conclude that the fluctuations did not originate from blinking by the non-radiative triplet state

of the fluorophore. The fluctuations could also not be explained by large distance changes between BtuCD and BtuF (visible as FRET) alone because most of the events were not characterised by anti-correlation of donor and acceptor intensities, but rather by changes in total intensity of both donor and acceptor combined (Supplementary Fig. 10). Similar changes in total intensity also occured in the quenching experiments with a single fluorescent label after addition of ATP (Fig. 1d). The fluctuations were absent when inspecting fluorescence traces of the ATPase inactive mutant BtuCD$_{EQ}$ (Fig. 1e). We reason that intensity fluctuations on the donor dye arise from ATP hydrolysis as described above. Experiments with unlabelled BtuCD$_{cys}$ and labelled BtuF indeed showed the absence of fluctuations in the fluorophore on BtuF (Supplementary Fig. 5). We thus conclude that the intensity fluctuations are transferred from the

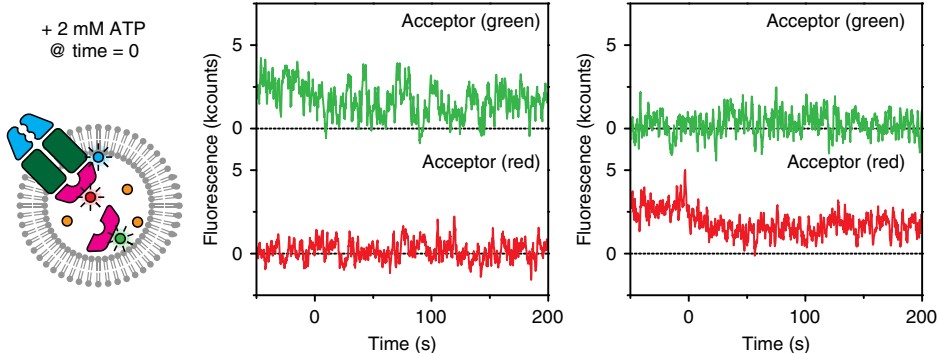

**Fig. 3** Absence of exchange observed with FRET. BtuCD$_{cys}$ labelled with Alexa Fluor 488 was reconstituted in liposomes loaded with 50% BtuF labelled with Alexa Fluor 555 (acceptor, green), 50% BtuF labelled with Alexa Fluor 647 (acceptor, red), and 100 μM vitamin B$_{12}$. One example of an initially bound 'green' BtuF (middle panel) and one of an initially bound 'red' BtuF (right panel) are displayed. While ATP and Mg$^{2+}$ were present, no increase in fluorescence intensity in the opposite channel (red or green respectively) was observed

donor dye on BtuCD$_{cys}$ to the acceptor dye on BtuF when bound to the complex (Fig. 2b), a conclusion that is also supported by data shown in Supplementary Fig. 9.

The dynamics that we attribute to conformational changes induced by ATP turnover took place on a timescale of seconds, which matches data obtained by ATPase rate experiments as described above (Fig. 1b: 15 ATP molecules per BtuCD-F complex per minute, not corrected for orientation and stoichiometry of two ATP molecules per BtuD sandwich dimer). The different conformations of the BtuCD-F complex possibly correspond to states in which BtuF either is capable of binding the substrate vitamin B$_{12}$ from solution—while still attached to BtuCD—or occludes the binding site for vitamin B$_{12}$. When we repeated this experiment with 100 μM vitamin B$_{12}$ loaded inside the lumen of the liposomes, a similar behaviour was observed as in the absence of substrate (Fig. 2c). These observations correlate well with our observations that show ATPase activity is insensitive to vitamin B$_{12}$ concentrations (Fig. 1b).

**Stability of the BtuCD-F complex during ATP hydrolysis.** Next, we used three-colour labelling in combination with binary FRET to investigate further the binding of BtuF to BtuCD: is the complex dissociating at the fast timescale of ATPase activity? BtuCD$_{cys}$ was labelled with donor fluorophore Alexa Fluor 488, while a 1:1 mixture of BtuF labelled with Alexa Fluor 555 (acceptor green) or Alexa Fluor 647 (acceptor red) was loaded into the liposomes together with 100 μM vitamin B$_{12}$ (Fig. 3, left panel). Using this strategy, initially one of the two acceptor colours is visible upon donor excitation as a stable complex is present. When ATP is added and BtuF would be exchanged, half of these exchange events would correspond to swapping between a 'red' and 'green' protein and be visible as a change in acceptor colour. In this binary experiment—i.e. we are only interested in whether a 'red' or 'green' BtuF is bound or not—we used liposomes with a diameter of 200 nm to increase the number of BtuF molecules in the lumen (8 instead of 1) at the same concentration (3.2 μM) as the previous experiments that used liposomes with 100-nm diameter; we selected only liposomes in which both BtuF colours were present (as detected by direct acceptor excitation). In the presence of ATP, we exclusively observed FRET traces showing a sustained interaction between BtuC and either the red- or green-labelled BtuF (Fig. 3, middle and right panel). By visual inspection, none of the several hundreds of complexes observed showed exchange of BtuF, confirming a model in which BtuF is stably bound to the transporter during ATPase activity with exchange rarely taking place on a timescale of minutes only

(Supplementary Fig. 9). Our findings are strongly supported by previous ensemble characterizations of the transporter embedded in liposomes where the authors looked at exchange of labelled versus unlabelled BtuF and no exchange was observed[16].

**Transport of single vitamin B$_{12}$ molecules.** To gain direct insight into the transport of vitamin B$_{12}$, we employed an experimental design in which we made use of fluorescence quenching by the cobalt ion[25] in the vitamin B$_{12}$ molecule. For these experiments, we labelled BtuF with Alexa Fluor 488, a dye that is quenched strongly by vitamin B$_{12}$ (Supplementary Table 1). To demonstrate the phenomenon, we immobilised labelled and His-tagged BtuF on a glass surface via anti-His antibodies and then introduced vitamin B$_{12}$ (which binds to BtuF with a $K_d$ of 15 nM[26]). The observed 76% quenching of fluorescence (Fig. 4a) is close to the theoretically predicted 84% (Supplementary Table 1). We next performed experiments in the context of the full transporter: unlabelled BtuCD$_{cys}$ was reconstituted with labelled BtuF and 10 μM vitamin B$_{12}$ (~3 molecules) in the lumen. Addition of ATP on the outside triggered a decrease in fluorescence intensity in roughly 5% (or 11% when corrected for orientation) of analysed traces (Fig. 4b, middle panel), which was only observed when substrate was loaded into the liposomes (Supplementary Fig. 11). Vitamin B$_{12}$ molecules that are freely diffusing inside the lumen of the liposome are too far from the fluorophore to yield any quenching. This substrate quenching effect is clearly different from the quenching effect observed in Figs. 1 and 2 because the former was observed only with Alexa Fluor 488-labelled BtuF, and did not occur with Alexa Fluor 555 and 647 (Supplementary Table 1), whereas the latter effect was insensitive to substrate. The substrate-induced reduced intensity persisted for extended times (up to minutes) indicating the continuous presence of vitamin B$_{12}$ in the transport complex after initiating substrate binding by ATP addition which probably (partially) opened the complex. Binding of new vitamin B$_{12}$ molecules to the BtuCD-F complex after translocation of previous substrate molecules apparently took place on a time scale faster than our acquisition, as no temporary increase in fluorescence was observed. However, signal intensities rose again after some time, possibly indicating that all substrate molecules had been transported out of the lumen of the liposomes. When we repeated the experiment with 10 times higher concentrations of vitamin B$_{12}$, the fluorescence signal remained low after ATP introduction (Fig. 4c), consistent with the interpretation that the rise in intensity correlated with substrate depletion from the lumen. If the residence time of a single vitamin B$_{12}$ molecule is described by an exponentially

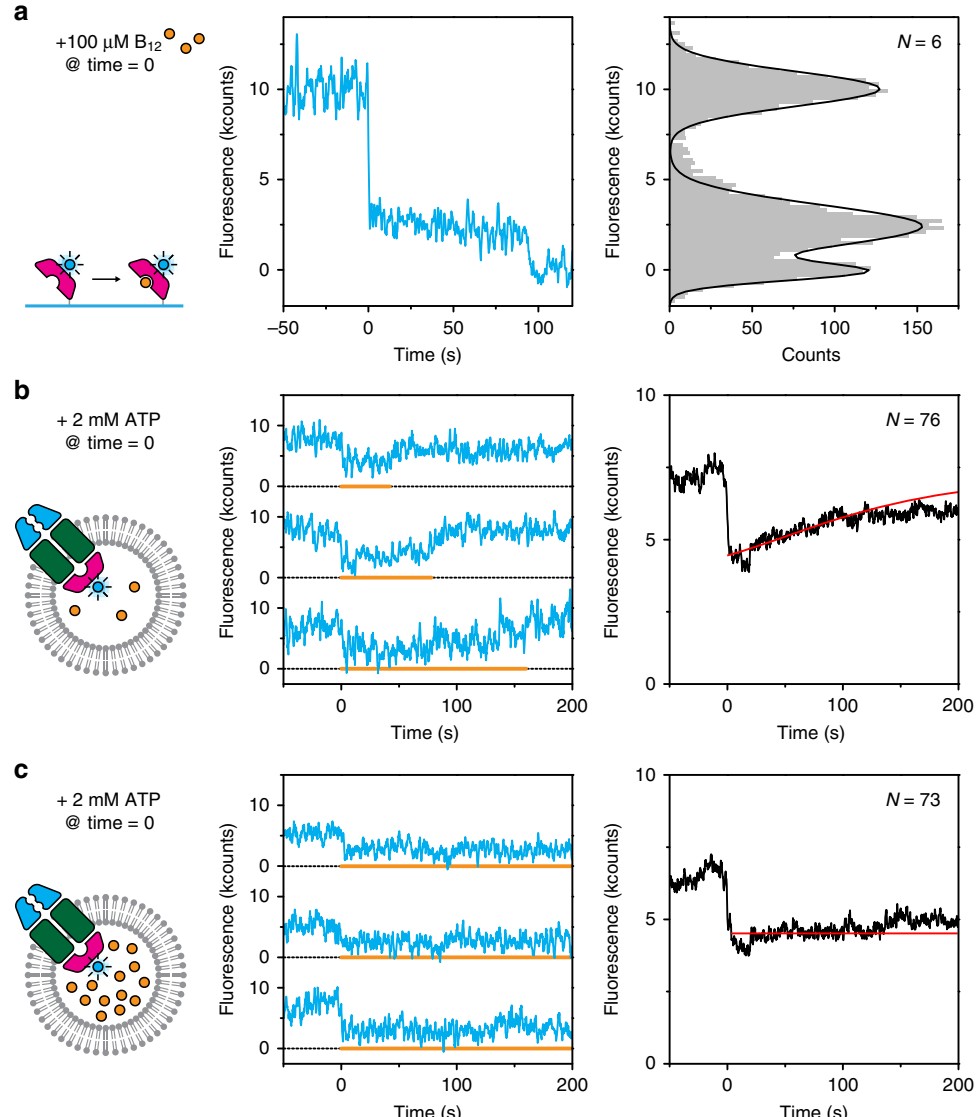

**Fig. 4** Transport of vitamin B₁₂ observed with fluorescence quenching. **a** Dye quenching by vitamin B₁₂. His-tagged BtuF labelled with Alexa Fluor 488 was immobilised on the glass surface via anti-His antibodies. Upon introduction of substrate, the fluorescence was quenched (middle panel) before it bleached around 100 s. Fluorescence intensities were collected in a histogram (right panel) to show that the quenching effect was ~75%. **b** When unlabelled BtuCD was reconstituted in liposomes loaded with 10 μM vitamin B₁₂ (~3 molecules) and BtuF labelled with Alexa Fluor 488, a decrease in intensity was observed upon introduction of ATP and Mg²⁺ (middle panel). The orange line marks the periods of low fluorescence and indicates when a substrate molecule was present in the transporter. Averaging all traces (right panel) shows a gradual increase in intensity, which we attribute to depletion of substrate from the liposome. For data analysis, see methods. **c** Similar experiment as described in **b**, but with ten times higher concentration of vitamin B₁₂. Depletion as observed in **b** was not visible here. For each condition, more than roughly 1000 traces were analysed

decreasing distribution, the distribution of total residence times for multiple molecules is given by a rise-and-decay function (see Methods). Since the number of substrate molecules inside a liposome follows a Poisson distribution, the total residence time of all vitamin molecules in a single transporter in one proteoliposome is described by a convolution of both aforementioned distributions. Summing all traces that show a decrease in fluorescence intensity upon addition of ATP, directly yields the cumulative convoluted distribution. We fitted this distribution and extracted an average residence time of $40 \pm 10$ s per substrate molecule in the transporter (Fig. 4b, right panel; Methods). Independent of substrate concentration, fluorescence traces of complexes that do not show an initial ATP-dependent quenching effect remain flat, indicating that those transporters are unable to translocate vitamin (Supplementary Fig. 11).

**Transport model.** Our study of the transport mechanism of BtuCD-F transporters embedded in lipid bilayers at the single-molecule level provides direct insight into the relation between BtuCD, BtuF and substrate transport. Our results build upon previously reported ensemble measurements and enable us to propose a transport model that unifies different existing models and biochemical and structural data (Fig. 5; corresponding crystal structures are noted in the figure legend). In the absence of nucleotides and substrate, a stable BtuCD-F complex is formed in the lipid bilayer (state I). We have shown that such a stable complex exists when the protein is embedded in a lipid bilayer. Binding and hydrolysis of ATP causes a conformational change in the complex, but does not lead to the dissociation of BtuF from the complex (state II) (Figs. 1b, d–f, 2 and 3). In the absence of nucleotides and substrate, BtuF fully associates with the complex

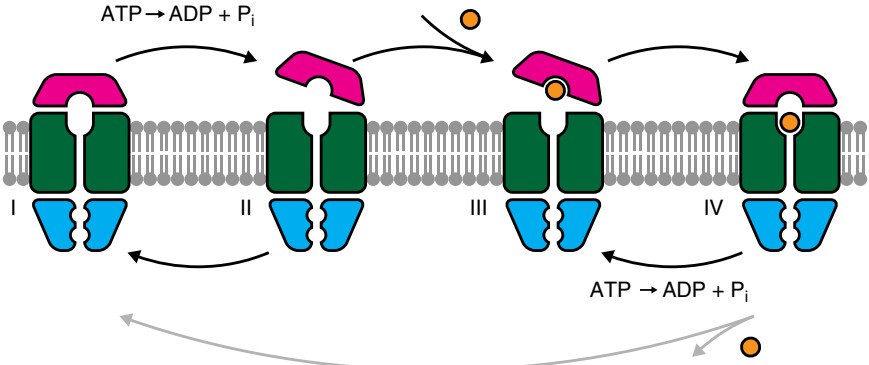

**Fig. 5** Transport model. BtuCD-F forms a stable complex in the ground state (I, corresponding to PDB ID 2QI9 and 4DBL). Binding and hydrolysis of ATP partially dislodges BtuF from the transporter (II, related to PDB ID 1L7V and 4R9U). In the absence of vitamin $B_{12}$, the complex cycles back and forth between states I and II. When substrate is present, BtuF can catch it (III), and the complex will fully associate capturing the vitamin inside the complex (IV, corresponding to PDB ID 4FI3). From this state, the transporter can either translocate the substrate molecule and return to the ground state (I), or it can return via ATP binding and hydrolysis back to the previous state (III). The latter pathway, which does not transport any molecule, is more likely to happen

again: this cycle between states I and II explains the high basal ATPase activity (Figs. 1b and 2b, c). We propose that the conformational change towards state II results in partially opening of the BtuCD-F complex allowing a vitamin $B_{12}$ molecule to bind in the high-affinity pocket of BtuF (state III). BtuF consequently seals off the periplasmic side of the transporter and the substrate molecule resides in the complex (state IV). The existence of such a state has recently been demonstrated by a crystal structure[15]. Our results also suggest that the substrate resides in a different place than the BtuF-binding pocket as the quenching effect in the full transporter is lower than in BtuF alone (Fig. 4a, b), although we cannot exclude that a photophysical effect reduced the fluorophore quenching in the full complex. Upon binding and hydrolysis of ATP, the complex returns to state III. The transporter can cycle multiple times between states III and IV without releasing vitamin $B_{12}$ (Fig. 4b, c), resulting in a much higher ATPase rate than transport rate (Fig. 1b and Supplementary Fig. 1). This cycling is equivalent to the cycling between states I and II in the absence of substrate. Occasionally, a vitamin $B_{12}$ molecule is released on the cytoplasmic side of the membrane via a proposed intermediate state[16]. When the binding location becomes vacant, a new vitamin $B_{12}$ molecule is bound rapidly. In conclusion, ATPase activity and transport are uncoupled and ATP hydrolysis is continuously required to (unproductively) reset the transporter.

## Discussion

In our model, we propose that the Btu-CD transporter shows ATPase activity while the vitamin $B_{12}$ substrate remains bound. In the absence of tools to simultaneously measure ATPase activity and substrate binding in single-molecule studies, we base our reasoning on observed time scales of ATPase and transport rate (of which the ATPase rate is at least 10-fold higher) and occurrence numbers: the percentage of complexes that is responsive to ATP (18%, Fig. 1) is similar to liposomes that show export of substrate upon addition of ATP (11%, Fig. 4), indicating we are looking at the same subset of complexes. Formally, it cannot be excluded that in Fig. 4 we probe a fraction of transporters—different from the fraction with high basal ATPase-activity fraction (Fig. 1)—that exhibits strong coupling between ATP turnover and transport. Since ATP turnover of this hypothetical coupled fraction would be at least hundred-fold slower than the fraction with high ATP hydrolysis activity, this fraction would be invisible in Fig. 1. However, this assumption would

mean that complexes showing ATP-induced transmembrane conformational changes are transport-inactive. Moreover, the observed response times of ATPase activity (Fig. 1) and ATP induces quenching of Alexa Fluor 488-labelled BtuF (Supplementary Fig. 11) are similar. Therefore, the most parsimonious interpretation is that the highly ATPase-active and transport-active fractions overlap, but alternative interpretations could affect conclusions regarding the ratio of number of ATP molecules hydrolysed per transported substrate. Nonetheless, our model is consistent with the observed low transport efficiency, the insensitivity of ATP hydrolysis to the presence of vitamin $B_{12}$, and the high basal ATPase activity[19]. It unifies aforementioned findings with structural data into a new transport model[12,14–16,27]. Importantly, we observed that BtuF remains associated with BtuCD for extended times. Borths et al. also proposed that BtuF might remain bound to the transporter, although direct evidence was not available, and their assumption was based on observations of the complex in detergent solution rather than lipid bilayers. Korkhov et al.[16] also performed binding experiments of BtuF to BtuCD, however they did not draw conclusions about binding times and dislodging of BtuF from the TMDs in their model. In this work, we observed binding times that are longer than timescales of substrate transport and much longer than timescales of ATP hydrolysis. In previous studies, also the low transport efficiency and substrate entrance into the transporter remained unclear. Korkhov et al. incorporated a state in their model with vitamin $B_{12}$ locked inside the complex, based on crystal structures and ensemble studies in detergent. However, their model lacked a clear explanation for the inefficiency of coupling between ATP hydrolysis and transport. In our model, we directly show continuous turnover of ATP—with or without substrate. ATP binding and hydrolysis is required to continuously reset the transporter, i.e. induce structural changes that allow for substrate binding on the periplasmic site. Some of this energy might also induce structural changes that create a pathway for substrate escape to the cytosol, however, currently the structural evidence for that change is missing. As vitamin $B_{12}$ can be translocated against a concentration gradient, input of metabolic energy is required though[19]. An additional role of BtuF—besides capturing substrate with high affinity—is probably to close the periplasmic site of the transporter when a translocation channel is formed to the cytoplasm. Dissociation of BtuF from BtuCD is rare, and not an obligatory step for transport. Occasional dissociation may be related to escape from off-path conformations such as the asymmetric state with collapsed cavity trapped in a

crystal structure[27]. The futile energy consumption in the absence and even presence of substrate, also previously noted in the ABC exporter P-glycoprotein[28] and other ABC importers[9], might raise questions about the biological purpose. Although the basal ATPase activity in the P-glycoprotein might be explained by the large variety of substrates it can transport, only two substrates are known to be imported by BtuCD-F: cobalamin (vitamin $B_{12}$) and the chemically related cobinamide[29]. Therefore, the basal ATPase activity must be explained by other mechanisms. We speculate that this is a mechanism to effectively capture and transport rare essential substrates such as some B vitamins. Type I ABC importers for highly abundant nutrients, such as the maltose transporter MalFGK, seem to show tight coupling between ATP hydrolysis and substrate transport[21]. Possibly there is also no evolutionary drive to optimise transporters for rare substrates as total ATP consumption for these transporters is negligible compared to total cellular ATP consumption.

While our model unifies the currently available ensemble and single-molecule data, there are still questions remaining: is the ATPase activity originating only from transport-active complexes? A dose-dependent relation between ATPase activity and substrate transport could give insight. What is the structural basis of the BtuCD-F complex that is susceptible to vitamin $B_{12}$ binding (states II and III)? What is the escape path for substrate translocation? And how much futile ATP hydrolysis takes place in the substrate-free and bound state of the complex during each complete transport cycle? Further structural and biochemical characterisation on ensemble and single-molecule level is needed to understand the transporter in more detail.

## Methods

**Purification and labelling of BtuCD-F.** BtuC (N-terminal octa-histidine tag (His$_8$ tag)) and BtuD, both with all native cysteines replaced by serines (referred to as cys-less), were cloned into the dual-expression vector p2BAD[30]. Additional mutations Q111C and W115L (Supplementary Table 2) were introduced in BtuC (referred to as BtuCD$_{cys}$). An additional mutation in the Walker-B motif (E159Q) of BtuD was introduced to impair ATPase activity (referred to as BtuCD$_{EQ}$). BtuCD was expressed in standard *Escherichia coli* BL21 (New England Biolabs). Cells where grown in LB broth (Miller) + 100 µg ml$^{-1}$ ampicillin and upon reaching stationary phase induced for 1 h with 0.01 % arabinose. Cells were collected by centrifugation (20 min, 5000 RPM, 4 °C) and resuspended in 50 mM KPi. Membrane vesicles were prepared immediately and stored at −80 °C after freezing in liquid nitrogen. Stored membrane vesicles were solubilized in 1% dodecyl maltoside (DDM, Anatrace) for one hour and insoluble material was removed by centrifugation (20 min, 80,000 RPM). Buffers used were 50 mM KPi, 200 mM KCl, pH 7.5 (buffer A) + 0.03% DDM (buffer B). Clear supernatant was added to 500 µl (bed volume) of Ni Sepharose (GE Healthcare) pre-washed in buffer B. The mixture was incubated for 1 h and washed with buffer B + 50 mM imidazole to remove non-specifically bound proteins. While immobilised on the nickel column, BtuCD was labelled with the appropriate dye (Alexa Fluor 488, Alexa Fluor 555 or Alexa Fluor 647, Thermo Fisher Scientific) at a concentration of 100 µg ml$^{-1}$ in buffer B at 4 °C for 15 min while gently mixing. Reaction volumes were 500 µl + bed volume. With BtuC being a homodimer in the complex, we aimed for 1 label per complex. Actual yields varied between—on average—0.5 and 1.5 labels per complex (see below for determination). Dual-labelled BtuCD cannot be excluded here, but are identified and dealt with during data analysis (see below). Free dye was removed by washing with 20 column volumes of buffer B. Labelled BtuCD was eluted with buffer B + 500 mM imidazole. Protein was further purified by using size-exclusion chromatography (SEC, Superdex-200 column, GE Healthcare) and directly used for reconstitution into liposomes. Protein yields and labelling efficiencies were determined with SEC by measuring absorption at 280 nm and the appropriate dye wavelength. BtuF (N-terminal His$_8$ tag, signal peptide removed, cys-less) was cloned into pBAD18 with additional mutation D141C. The protein was expressed and purified separately from BtuCD with the protocol as described above, with the omission of DDM and an increased dye labelling incubation time of 2 h.

**Reconstitution into proteoliposomes.** Empty liposomes were prepared from a synthetic lipid mixture of 40% (w/w) 1,2-dioleoyl-sn-glycero-3-phosphocholine (DOPC), 29% (w/w) 1, 2-dioleoyl-sn-glycero-3-phosphoethanolamine (DOPE), 30% (w/w) 1,2-dioleoyl-sn-glycero-3-phospho-(19-rac-glycerol) (DOPG) and 1% (w/w) 1,2-dioleoyl-sn-glycero-3-phosphoethanolamine-N-(cap biotinyl) (biotin-DOPE) in buffer A and extruded at 400 nm (Mini-Extruder with polycarbonate filter, Avanti Polar Lipids) after being subjected to four freeze-thaw cycles (frozen

in liquid nitrogen and thawed at room temperate)[31]. For reconstitution, empty liposomes (10 mg ml$^{-1}$), BtuF (if applicable, 3.2 µM), Triton X-100 (Sigma-Aldrich, 0.3%) and BtuCD (0.02 µM) were mixed in buffer A. For 100-nm liposomes, above concentrations lead to 10% of the liposomes having 1 BtuCD molecule reconstituted, and only 0.5% having more than 1 molecule; on average 1 BtuF molecule is found inside a vesicle. In case liposomes of different size were used, concentrations were adjusted to maintain the BtuCD / liposome ratio. BioBeads SM2 (BioRad, 40 mg ml$^{-1}$ mixture) were added five times to remove detergent for the following incubation periods at 4 °C while gently agitated: 15 min, 15 min, 30 min, overnight, and 60 min. BioBeads were then removed and the solution stored in liquid nitrogen.

In order to incorporate additional molecules such as vitamin $B_{12}$ or ATP (Sigma-Aldrich), where applicable, proteoliposomes were mixed with these components. Next, the solution was subjected to four freeze-thaw cycles to homogenise the sample and subsequently proteoliposomes were extruded at 100 nm (this diameter is always used, unless explicitly stated otherwise). Dynamic light scattering measurements on these samples show an outer diameter of 130 nm with a polydispersity of 15%; liposomes extruded at 200 nm measure a diameter of 160 nm with a polydispersity of 17%. In case BtuF was used throughout the procedure, excess extraluminal protein was removed by incubating the solution with Ni-NTA magnetic agarose beads (Qiagen) for 1 h in a reaction tube and extracting them with a magnet.

**ATPase activity assay.** Proteoliposomes were prepared as described above, with a mass ratio BtuCD:lipids of 5:800. If applicable, BtuF was co-reconstituted at 3 µM and vitamin $B_{12}$ was incorporated at 100 µM. All samples were subjected to four freeze-thaw cycles and subsequently extruded at 100 nm.

ATPase activity was quantified using an NADH-coupled enzyme reaction with pyruvate kinase and lactic dehydrogenase (PK/LDH). Reaction mixtures of 300 µl, containing proteoliposomes (loaded with BtuF and vitamin $B_{12}$ if applicable) with a final concentration of 180 nM BtuCD, 4 mM phosphoenol pyruvate (PEP, Sigma-Aldrich), 0.3 mM NADH (Sigma-Aldrich) and four units of PK/LDH mix (Sigma-Aldrich) in buffer A, were prepared freshly. All mixtures were pre-incubated for 3 min at 30 °C. Reactions were initiated by adding 2 mM ATP (saturating concentration[19]) and 10 mM MgCl$_2$ and also took place at 30 °C. Formation of ADP was continuously monitored by measuring light absorption at 340 nm.

**Single-molecule fluorescence microscopy.** Flow cells were constructed as described in Supplementary Fig. 12 to allow for a fast exchange in buffer with a steep gradient. Functionalization of the glass surface is adapted from Tabaei et al.[32]. Microscope cover glass (no. 1.5) was extensively cleaned by 20 min sonication in isopropanol, followed by 20 min sonication in acetone (after rinsing with ultrapure (Type I) water), followed by 10 min plasma cleaning of the dried glasses. After flow cells were constructed, they were incubated with 1 M KOH for 30 min. Next, the channel was washed with copious amounts of ultrapure water and then washed with PBS buffer. A mixture of 1 mg ml$^{-1}$ polylysine (20 kDa) grafted with polyethylene glycol (PLL-PEG) (PLL(20)-g[3.5]-PEG(2), SuSoS) and 2 µg ml$^{-1}$ PLL-PEG-biotin (PLL(20)-g[3.5]-PEG(2)/PEG(3.4)-biotin(20%), SuSoS) was incubated for 60 min. Finally the channel was washed again in PBS buffer.

Prior to use, the flow channel was incubated with 1.0 mg ml$^{-1}$ bovine serum albumin (BSA) in PBS (buffer C) for 5 min. Following this step, 0.1 mg ml$^{-1}$ streptavidin (streptavidin from *Streptomyces avidinii*, Sigma-Aldrich) in buffer C was incubated for 5 min. Subsequently, unbound streptavidin was washed out with buffer C. Next, fresh proteoliposomes diluted to 100 µg ml$^{-1}$ (1.5 nM liposomes) in buffer A were introduced to the flow cell and incubated for 10 min. Freshly degassed buffer A with the GODCAT oxygen scavenging system[33] (7.5 U ml$^{-1}$ glucose oxidase (glucose oxidase from *Aspergillus niger* type VII, Sigma-Aldrich), 450 U ml$^{-1}$ catalase (catalase from bovine liver, Sigma-Aldrich) and 0.8% (w/v) glucose monohydrate) (preparation buffer) was used to wash away unbound proteoliposomes. During image acquisition, activation buffer was introduced into the flow channel (Supplementary Fig. 12). Activation buffer equals preparation buffer plus (depending on the type of experiment) 2 mM ATP (saturating concentration) and 10 mM MgCl$_2$, or BtuF. A programmable syringe pump (NE-1000, New Era Pump Systems, Inc.) precisely controls all flows. In case BtuF was directly immobilised on the glass surface, the protocol is slightly altered: instead of flowing proteoliposomes, 1 µg ml$^{-1}$ anti His-tag antibodies with biotin conjugate (Thermo Fisher Scientific) were incubated for 10 min. Next, unbound antibodies were washed out with buffer A and subsequently 1 nM BtuF was incubated for 5 min. Following this, the protocol is continued with introducing preparation buffer. Activation buffer equals preparation buffer plus 2 µM vitamin $B_{12}$. All experiments were performed at room temperature.

Using total internal reflection fluorescence (TIRF) microscopy, proteoliposomes were visualised with an Olympus IX–71 inverted microscope using a ×100 high-numerical-aperture TIRF objective (Olympus UApo N 100× O TIRF). Dyes were excited with 488, 532, or 637-nm lasers at ~10 W cm$^{-2}$. Fluorescence was recorded with an EM-CCD camera (Hamamatsu Photonics) at highest EM-gain and a frame rate of 5 Hz. In case of measuring fluorescence resonance energy transfer (FRET), fluorescence signals were separated by a two-channel simultaneous imaging system (DV2, Photometrics) and projected on the camera. Each FRET acquisition was preceded and followed by a 5 s period of acceptor only excitation. Emission

intensities of donor and acceptor molecules allowed for information about the number of dyes.

**Data analysis**. In-house written software for ImageJ[34] was used to first correct acquired images for laser beam profile and electronic offset from the camera. The software continues to identify peaks in one channel (for FRET data it also finds corresponding spots in the other channel) using built-in functions of ImageJ. Fluorescence traces were extracted by integrating the fluorescence counts in a circle with 7 pixel diameter. Integrated background counts, normalised to a circle with 7 pixel diameter and extracted from a nearby area devoid of peaks, were smoothened by a temporal median filter of 6 s and subtracted from the trace. Before continuing, resulting traces were subjected to a temporal mean filter of 1 s to remove noise. If applicable, traces were finally corrected for cross talk between the different fluorescence channels.

Changes in fluorescence transients were detected by identifying a decrease or increase at neighbouring time points that exceeds a certain threshold value. The threshold is set to the standard deviation of the signal in the 50 s before $t = 0$, times a constant, typically 3.5. A histogram was created for increase or decrease events. A positive signal (a peak) in the histogram was fitted with a convolution of the system response function $R(t) = A_r \exp\left(-\frac{t^2}{2\sigma^2}\right)$ with width $\sigma$ and scaling constant $A$, and an exponential decay function $S(t>0) = A_s \exp\left(-\frac{t}{\tau}\right)$ with decay time $\tau$:

$$R * S(t) = A \exp\left(\frac{\sigma^2 - 2\tau t}{2\tau^2}\right)\left(1 - \text{erf}\left(\frac{\sigma^2 - \tau t}{\sqrt{2}\sigma\tau}\right)\right).$$

To analyse the residence time $\tau$ of multiple vitamin $B_{12}$ molecules in a set of liposomes, the following equations were used: first single molecules were assumed to have a residence time $\tau$ distribution that follows a single exponential decay. The probability distribution $p(t)_{\tau,N}$ of the total residence time of $N$ substrate molecules inside a liposome can then be modelled with a rise and decay function[35]: $p(t)_{\tau,N} = \frac{\tau^{-N}t^{N-1}}{(N-1)!}\exp\left(-\frac{t}{\tau}\right)$. The relative probability of the expected number of molecules inside a liposome is described by the Poisson distribution $p_N = \frac{\overline{N}^N \exp(-\overline{N})}{N!}$, where $\overline{N}$ equals the average number of molecules in a liposome. Multiplying both probability distributions and summing over all positive $N$ gives the probability distribution of the total residence time in a collection of equally sized liposomes: $p(t)_\tau = \sum_{N=1}^{\infty} p(t)_{\tau,N} p_N$. The cumulative distribution function is used to fit the data: $\int_{t'=0}^{t} p(t')_\tau$.

**Data availability**. The data that support the findings of this study are available from the corresponding authors upon reasonable request.

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

## Acknowledgements

We thank C.M. Punter for his help with developing software, V.V. Krasnikov for his help with the design, construction and maintenance of the single-molecule fluorescence microscopes, S. Rempel for his support on radiolabelled substrate transport experiments, T. Pijning for dynamic light scattering experiments, and G.B. Erkens for sharing experience on molecular biology and single-molecule imaging of membrane proteins. A. M.v.O. acknowledges funding from the European Research Council (ERC Starting grant 281098; SINGLEREPLISOME) and the Netherlands Organisation for Scientific Research (NWO Vici grant 680–47–607). D.J.S. acknowledges funding from the Netherlands Organisation for Scientific Research (NWO) (Vici grant 865.11.001) and the European Research Council (ERC) (ERC Starting Grant 282083).

## Author contributions

J.M.H.G. performed experiments; J.M.H.G., D.J.S. and A.M.v.O. designed experiments, wrote the manuscript and contributed to the interpretation of the data.

## Additional information

**Competing interests:** The authors declare no competing financial interests.

