## [Peer Review File · Nature Communications]

Reviewers' comments:

Reviewer #1 (Remarks to the Author):

The type II ABC importer BtuC2D2-BtuF catalyzes the uptake of vitamin B12 into bacteria. The structure and function of high-affinity vitamin binding protein BtuF as well as different states of the full ABC import complex have been characterized. Despite the fact that this work has been published in high-rank journals, a number of questions still remain and some surprising observations need to be addressed by novel approaches including an optimized reconstitution system and lipid micro-environment.

Goudsmits and colleagues used single molecule techniques to indirectly analyze vitamin B12 transport by BtuCD-F. Different steps could be followed, applying single-molecule fluorescence quenching and smFRET on BtuCD-BtuF complexes reconstituted in a diluted stochastic manner. Based on these results, it seems that the high-affinity binding protein BtuF stays bound to the transport complex BtuCD for several rounds of non-productive ATPase hydrolysis. Moreover, futile ATP hydrolysis is even observed in the vitamin B12 loaded state.

The experiments are well designed and comprehensively presented. Since other models and data, in which BtuF dissociates during the transport process are published, the authors should discuss the discrepancy in more details. In addition, some controls and careful statistics need to be included for each experiment as only a few –most likely the best– traces are shown. The study would profit from important controls in order to strengthen the conclusions and working model (see below). In addition, it is essential to include direct transport and ATPase measurements under similar conditions as well as ATPase inactive variants to confirm that the fluorescence fluctuations are indeed related to ATP hydrolysis and occasionally to translocation events.

Major critical points:

- 1) The fluorescence quenching induced by ATP hydrolysis only indicates a structural change of the environment of the fluorophore. This experiment does not allow to judge whether the binding to BtuF is influenced. This should be clarified.
- 2) BtuC2D2 is a symmetric homodimer, which becomes an asymmetric complex upon BtuF binding (Korkhov et al., 2012). However, schemes suggest that BtuC dimer is labeled sub-stoichiometrically. So different results are expected if one of the two subunits or all two subunits are labeled. The authors should present data for a 1C':1, 1C'':1, and 2C:1 labeling of BtuC2D2 and compare the outcome. Solid statistics are critical for the interpretation of the data.
- 3) The title "single molecule visualization of transport" might lead to misunderstandings as no direct transport or ATP hydrolysis events were traced. The title should be changed and focus on the finding that unproductive cycles of ATPase in the apo and substrate bound state of BtuCD were observed. Fluorescence fluctuations in fluorescence are interpreted as conformational changes or B12 release from BtuF, but not as direct membrane translocation or ATP hydrolysis events.
- 4) Since the fluorescence data only indirectly report on ATP hydrolysis and vitamin B12 transport, the general readership would highly appreciate if the turnover rates of ATP and B12 are directly measured simply by radiotracers and compared under similar experimental conditions. This again is a very critical point and requires a concise interpretation of the fluorescence data.
- 5) In Figure 2 the dynamic fluctuations are interpreted as ATP hydrolysis events. Why do such events also occur before ATP is added? From visual inspection I cannot see more fluctuations in Fig. 2b/c than in Fig. 2a. Can the authors show a zoom-in of these fluctuations and quantify them? The number of events should be dependent on ATP concentration. Such a study would strengthen

the chapter dealing with the fluctuations as well as the interpretation of the data.

6) The study rests on the very low transport efficiency and high basal ATPase activity, which is not affected by vitamin B12. However, it would be key to provide direct data for the stoichiometry of uncoupling for the system used under identical conditions (see above). It is essential to know how many liposomes containing BtuCD-BtuF in the proper orientation are transport active and which are not. This is crucial to interpret the highly uncoupled ATP and transport turnover rates.

7) Based on the Venus-Fly trap mechanics, the opening and closing of substrate-binding proteins can be elegantly followed by smFRET. The reader may ask why the authors did not apply these established approaches to probe the opening/closing of Btu-F in relation to the ATP turnover as well as in the presence and absence of vitamin B12.

8) It remains unclear why different sizes of liposomes were used for different topics. Please provide comments. In addition, it is important to experimentally confirm the diameter and the derivation by dynamic light scattering.

9) It is yet ambiguous whether the intensity fluctuations really reflect on ATP hydrolysis. In order to support this idea, the authors need to analyze hydrolysis inactive mutants such as the E-to-Q substitution of the catalytic base. Those need to be included to strengthen the overall conclusions of the manuscript.

Specific points:

1) Important details of the experiments (ATP concentration, liposome, size etc.) should be included in each figure legend to simplify the understanding.

2) Can the authors provide more details about the transporter labelling and the quantification of the labelling efficiency?

3) What is ultrapure water?

4) Which objective was really used? Single-molecule experts would like to know these details. It is unclear why a 6-s filter was used. Please comment.

5) Figure 1: The fluorophore is already present at BtuF before the incubation. This should therefore be included in the left panel of the illustration.

6) Legend of Fig. 1: it would read "marked with a yellow dot" instead of red dot.

7) Legend of Supplemental Figure 3: it should read "marked with a black line" instead of red line.

8) Supplemental Figure 9: The authors need to explain why the fluorescence does not stay constant after buffer exchange using their new setup of the flow-chamber (red curve). Which fluorescence molecules are detected?

9) Supplemental Figure 7 shows strong fluctuations of another donor molecule with blinking events on the left. The same appears at the acceptor trace (bottom right). The authors need to comment on blinking and FRET events.

10) Supplemental Figure 1: Details of the transport conditions must be included into the legend.

11) Figure 2 and 3: FRET traces should be shown. This would help the reader to differentiate between correlation and no correlation.

12) Figure 2: what is the correlation between the donor/acceptor traces to the figure b, right part on the right with the sum up?

13) Figure 2: the significance of the fluctuations is unclear. ~20% offset from the initial signal and the fluctuations rise often to the initial values.

14) Figure 1: How many molecules are analyzed in each of these histograms? What is the explanation for the rise of the signal in Fig. 1c?

Reviewer #2 (Remarks to the Author):

In this interesting manuscript, Goudsmits and coworkers report clever experiments at the single transporter level on the vitamin B12 importer BtuCD. This is very exciting work that provides novel insights into the working mechanism of a membrane transporter. The manuscript would benefit from further clarifications. The data also deserve better integration into what is currently known at the biochemical and cell biological level about BtuCD-F in relation to other ABC importers. A few key points are provided below.

1. In the summary, the BtuCD-F complex could be better introduced. The function of BtuF is not explained.

2. In the Introduction the difference in structural and functional properties of the periplasmic binding protein deserve attention; in contrast to other binding proteins, BtuF might not show substrate binding via a venus flytrap mechanism, and this could have important implications for how substrate binding to BtuF is linked to interactions of liganded and unliganded BtuF to BtuCD. Another element that deserves further explanation is the differences in the degree of substrate stimulation of the ATPase activity between different binding protein-dependent ABC importers, and the position of BtuCD-F in this. The information should not only excite the expert but also inform the interested reader about the importance of what was found. The Introduction finishes with the question whether "ATP is merely required to reset the transporter", but a clear answer regarding this point is missing at the end of the Discussion.

3. The text frequently refers to consistency between the current data and "ensemble measurements", "previous bulk measurements", "previous ensemble characterizations" etc., but the details of what is exactly similar, is left to be discovered by the reader; it is the authors who should explain and emphasize this in sufficient detail.

Results

4. p. 5: "While the fluorescence was high in the absence of nucleotide, in a fraction of liposomes (8%) it decreased rapidly upon addition of ATP to the solution surrounding the liposomes." The author subsequently state that this represents the fraction of active transporters. What is meant with "active" here? Given the known ATP concentration and known binding constant of BtuCD for ATP is should be possible to calculate the fractional occupation of ATP binding sites by ATP. Given the experimental conditions, this number will most likely be greater than 8%. What happens with the other 92%?

5. p 7: "The fluctuations could not be explained by FRET alone because many of the events were not characterized by anti-correlation of donor and acceptor intensities." Interesting statement but not clear.

6. p. 9: "After summing all traces that show a decrease in fluorescence intensity at zero seconds,

we extracted an average residence time of 40 +/- 10 seconds per substrate molecule in the transporter". This is a key conclusion in the paper, but this statement and explanation are really too short.

Discussion.

7. The authors state: "Although it is currently not possible to simultaneously measure ATPase activity and substrate presence in single-molecule studies" [.....]. However, as the main conclusion of this paper is based on a comparison of the residence time for vitamin B12 binding versus rate of ATP hydrolysis, it is vital to measure in the current test system with current experimental conditions what the rate of ATP hydrolysis is, even if this would be measured non-simultaneously and in bulk.

8. What is the role of the Btu-F binding protein in transport? Would it be possible for the ligand to directly bind to and dissociate from BtuCD without involvement of BtuF, and could this explain the high rate of ATP hydrolysis compared to the rate of transmembrane transport of the substrate?

9. What is the role of ATP binding and hydrolysis in transport by BtuCD-F? Does ATP hydrolysis occur in response to ligand binding to BtuCD? If ATP would merely reset the transporter to complete the transport cycle, then what drives the translocation process? Is transport of B12 directed down a concentration gradient into cells that utilize this ligand in metabolism, or uphill in a reaction that would require input of metabolic energy? Some discussion would be useful here.

10. "In our model, dissociation of BtuF from BtuCD is rare, and not an obligatory step for transport." How is this interaction structurally maintained; is there any information available from crystal structures regarding this point?

Reviewer #3 (Remarks to the Author):

Goudsmits et al. performed single-molecule imaging experiments to gain a deeper understanding of the transport mechanism in the VitB12 importer BtuCD-F, a type II class ABC transporter. BtuCD-F from *E. coli* is a well-studied model system and has been extensively characterized both structurally and biochemically. However, the authors indicate that a number of open questions remain which pertain to the order of events in the transport cycle and whether VitB12 transport is tightly coupled to ATP hydrolysis that must be answered in order to achieve an understanding of the transport (import) mechanism. These biological questions are certainly interesting and a single-molecule approach seems like an appropriate application to probe the types of questions that the authors would like to address.

After carefully reviewing the written manuscript, which is concisely written and appears to be appropriately referenced, my sense is that there are considerable shortcomings in the execution and interpretation of the experiments that currently preclude this manuscript from publication. Aside from the technical issues discussed below, one of the main conclusions presented, namely that BtuCDF non-productively hydrolyzes ATP even in the absence of B12, is somewhat troubling. While such futile energy consumption has been previously noted, the biological purpose for such a mechanism is entirely unclear?

Perhaps most importantly, the authors state at the onset that they are working with a preparation of protein that has a remarkably low active fraction: as stated on page 5, only 8% of the protein is active. While such low activity may be a good argument for performing single-molecule investigations (heterogeneous ensemble), this limitation seems to significantly compromise all downstream experiments as the statistics resulting from the relatively low number of molecules per imaging experiment appear lacking. In some instances, it would appear that at least some fraction of the conclusions may have been drawn from just a few carefully-chosen traces (for example, single traces in Fig. 2 are used to support stable BtuF binding in the absence of ATP, and quenching with FRET in the presence of ATP). These considerations, and the absence of

complementary bulk activity assays, aside from showing that the protein has some level of transport function, and/or ensemble fluorescence measurements of the specific activities described (such as ligand binding studies), make it extremely difficult to evaluate the robustness of the conclusions drawn from individual experiments. An additional complicating feature of the experiments described is shown in Figure 1, which shows that the "activities" defined in the present study are based on relatively small changes in fluorescence intensity for which there is no physical basis for interpretation. Are the photophysical properties of Alexa 555 well characterized? Why not use a donor fluorophore for which there is literature regarding the basis of fluorescence changes (ie. Cy3). What causes fluorescence to decrease in Alexa555 in the absence of vitamin B12 (Fig. 1c); what causes fluorescence to increase in the absence of B12 (Fig. 1e)? Can the authors show that one molecule can exhibit both effects in a biologically regulated manner? Are the observed changes in fluorescence intensity mitigated by mutations in the protein that block transport activity? In summary, these shortcomings of the data severely limits the impact of the work and appears to call into question the reliability of the conclusions.

Specific comments:

Figure 1: Interaction between BtuCD and BtuF in absence of B12 substrate, in the presence of ATP: The quenching of single, Alexa 555 on reconstituted BtuCD was shown to occur upon addition of ATP outside the proteoliposomes, and was dependent on the presence of BtuF, and apparently occurs only upon ATP hydrolysis as quenching does not occur upon the addition of AMP-PNP.

It is not explained why the specific labeling sites were picked, in particular for BtuCD where the quenching of fluorescence is argued to be due to a conformational change.

It is difficult to determine where the quenching step occurs in the sample traces shown as there are numerous fluctuations in fluorescence intensity. The number of single molecules (N) analyzed in the histograms on the right are not shown and the methods used to generate these histograms are not clearly described and controls are not delineated. Were the traces analyzed by eye or was there some idealization of the data?

Figure 2: FRET between BtuCD and BtuF: High FRET is observed when BtuCD-F complex formed: In the absence of B12 you see a drop in acceptor fluorescence upon addition of ATP, but also a drop in donor fluorescence, apparently due to quenching as in the first experiment.

It is difficult to make strong conclusions about this experiment in general given the number of confounding variables. Qualitatively it is difficult to tell the differences or similarities between the single traces in 2b and 2c, but there seems to be more dynamics in donor fluorescence in presence of B12. What are the fluctuations in Alexa 647 intensity?

Figure 3: Stability of BtuCD-F complex: If there is an exchange between two different BtuF molecules that are bound to BtuCD, there would be an observed FRET change since there are two BtuF populations in the liposomes, each labeled with a different dye.

No change in FRET can be discerned in the two traces shown. Some sort of ensemble statistics is absolutely necessary. Were any complexes observed on the 'blue' Alexa 488 channel? If not, why?

Figure 4: B12 Transport Assay.

Clear quenching of fluorescence is observed for BtuF alone in presence of B12 (Fig. 4a). As this appears to be the 'loudest' signal in the manuscript, additional analyses of this effect would seem reasonable such as a binding curve, or waiting time distribution, to show its concentration dependence. Given that B12 has a 15 nM affinity for BtuF (page 9) does this mean that all experiments in the paper containing B12 have this quenched fluorescence baseline?

Given this is nearly an 80% change in fluorescence intensity, it is difficult to understand the relative scale of the transport data. Even a single B12 molecule would quench the sensor (Kd: 15 nM, single molecule of B12 in lipo $\sim 3 \mu\text{M}$), and without an experiment where no B12 is added to the inside of the liposomes to compare with, it would appear that depletion of internal B12 is not complete by 200 seconds (Fig. 4b. right). Why is this the case? Shouldn't the fluorescence intensity ultimately achieve an 80% higher intensity? How does this signal change when mutations are made in the active site that block B12 binding or transport? Such controls seem essential to verify the nature of the signal being measured.

In Sup.Fig.7, a single trace shows periods of loss of acceptor fluorescence. The authors interpret this as unlabeled BtuF substituting for labeled BtuF. This behavior is informative if true, but could also be explained by photoblinking. Clear evidence must be provided to exclude such an

interpretation. Alternating laser excitation may be of use here. (The second trace in this figure is clearly and admittedly aberrant and should not have been shown.)

Reviewer #1 (Remarks to the Author):

The type II ABC importer BtuC2D2-BtuF catalyzes the uptake of vitamin B12 into bacteria. The structure and function of high-affinity vitamin binding protein BtuF as well as different states of the full ABC import complex have been characterized. Despite the fact that this work has been published in high-rank journals, a number of questions still remain and some surprising observations need to be addressed by novel approaches including an optimized reconstitution system and lipid micro-environment.

Goudsmits and colleagues used single molecule techniques to indirectly analyze vitamin B12 transport by BtuCD-F. Different steps could be followed, applying single-molecule fluorescence quenching and smFRET on BtuCD-BtuF complexes reconstituted in a diluted stochastic manner. Based on these results, it seems that the high-affinity binding protein BtuF stays bound to the transport complex BtuCD for several rounds of non-productive ATPase hydrolysis. Moreover, futile ATP hydrolysis is even observed in the vitamin B12 loaded state.

The experiments are well designed and comprehensively presented. Since other models and data, in which BtuF dissociates during the transport process are published, the authors should discuss the discrepancy in more details. In addition, some controls and careful statistics need to be included for each experiment as only a few –most likely the best– traces are shown. The study would profit from important controls in order to strengthen the conclusions and working model (see below). In addition, it is essential to include direct transport and ATPase measurements under similar conditions as well as ATPase inactive variants to confirm that the fluorescence fluctuations are indeed related to ATP hydrolysis and occasionally to translocation events.

The reviewer raises valid critical points. We expanded the introduction to put our work in a better perspective, and we also elaborated on the differences and similarities between our transport model and existing models in the discussion. We included direct ATPase measurements for mutants used, as well as a newly-created ATPase inactive mutant. This new control mutant we also analysed with our single-molecule techniques. Below you can find a point-by-point discussion.

Major critical points:

1) The fluorescence quenching induced by ATP hydrolysis only indicates a structural change of the environment of the fluorophore. This experiment does not allow to judge whether the binding to BtuF is influenced. This should be clarified.

It is correct that quenching induced by ATP hydrolysis, as displayed in Fig. 1d, only hints for structural changes that induce an alteration of the local environment of the dye. We now explicitly state this in the text. With additional experiments (absence vs. presence of ATP and/or BtuF, Fig. 1f, Supplementary Fig. 5 and 7) we deduce how BtuF is involved in the fluorescence quenching:

“The quenching of the fluorescence suggests an alteration of the local protein environment of the dye and thus a conformational change of the complex (Fig. 1d, middle) [24]. Other dyes such as Cy3 and tetramethylrhodamine (TMR) manifest the same quenching behaviour (data not shown). The distribution of quenching events plotted in a histogram (Fig. 1d, right) revealed an ATP response time in our flow cell of 1.9 +/- 0.5 sec (see Methods). When we performed the same experiment in the absence of BtuF, no events were observed

(Supplementary Fig. 5), indicating that the observed decrease in intensity was mediated by interaction with BtuF.”

In the next paragraph we analyze how the binding to BtuF is influenced.

2) BtuC2D2 is a symmetric homodimer, which becomes an asymmetric complex upon BtuF binding (Korkhov et al., 2012). However, schemes suggest that BtuC dimer is labeled sub-stoichiometrically. So different results are expected if one of the two subunits or all two subunits are labeled. The authors should present data for a 1C':1, 1C'':1, and 2C:1 labeling of BtuC2D2 and compare the outcome. Solid statistics are critical for the interpretation of the data.

Complexed BtuCD-F does indeed form an asymmetric structure. (Hvorup *et al.*, 2007) However, this asymmetry mainly applies to transmembrane helix (TM) 5. We positioned our label on the periplasmic loop connecting TM3 and TM4, which is affected to a much lesser degree. We used the readout – quenching of fluorescence – to construct a histogram of intensity drops as measured after addition of ATP (Fig. 1d), see below. We observed a single distribution of events at the current resolution, meaning that we cannot discriminate between signal originating from 1C' or 1C'' labelling.

3) The title “single molecule visualization of transport” might lead to misunderstandings as no direct transport or ATP hydrolysis events were traced. The title should be changed and focus on the finding that unproductive cycles of ATPase in the apo and substrate bound state of BtuCD were observed. Fluorescence fluctuations in fluorescence are interpreted as conformational changes or B12 release from BtuF, but not as direct membrane translocation or ATP hydrolysis events.

We understand the concerns of the reviewer about this point, and we changed the title of the manuscript to:

“Single-molecule visualization of the vitamin B₁₂ transport mechanism of the ABC importer BtuCD-F”

By adding the word ‘mechanism’ we emphasize that we looked with single-molecule techniques at several different steps in the transport reaction – i.e. ATPase activity, BtuF

induced events and vitamin B12 transport. Now, we also specify this with an extra phrase in the introduction:

“Here we report the direct observation at the single-molecule level of ATP, vitamin B12 and BtuF-induced events in the transporter complex embedded in liposomes.”

However, we must note that we do see the transport of a single vitamin B12 molecule out of the liposome, as we look at a signal that depends on substrate concentration (Fig. 4), thus allowing to see depletion of substrate from the lumen.

“However, signal intensities rose again after some time, possibly indicating that all substrate molecules had been transported out of the lumen of the liposomes. When we repeated the experiment with 10 times higher concentrations of vitamin B₁₂, the fluorescence signal remained low after ATP introduction (Fig. 4c), consistent with the interpretation that the rise in intensity correlated with substrate depletion from the lumen.”

4) Since the fluorescence data only indirectly report on ATP hydrolysis and vitamin B12 transport, the general readership would highly appreciate if the turnover rates of ATP and B12 are directly measured simply by radiotracers and compared under similar experimental conditions. This again is a very critical point and requires a concise interpretation of the fluorescence data.

The reviewer addresses a very clear point here, which we agree on, and we included these measurements. Moreover, we included extra experiments to determine the orientation of the BtuCD protein in the liposomes to even better interpret rates.

We added a new Supplementary Figure (2) which presents an experiment that directly determines the orientation. We found that 55% is oriented right-side out; 45% is oriented inside out.

We also added a bar chart with ATPase rate of various mutants we use (Fig. 1b). These numbers match the timescale at which we see ATP-dependent fluctuations in single-molecule traces.

“The dynamics that we attribute to conformational changes induced by ATP turnover took place on a timescale of seconds, which matches data obtained by ATPase rate experiments as described above (Fig. 1b: 15 ATP molecules per BtuCD-F complex per minute, not corrected for orientation and stoichiometry of 2 ATP molecules per BtuD sandwich dimer).”

As for uptake rates with radio-labelled vitamin B12, Supplementary Figure 1 contains these data. From the legend:

“Uptake rates should not be considered as absolute numbers, because accumulated ADP might inhibit the transporter, there is a possibility of multi-lamellar 400-nm proteoliposomes and during the filtering step in the method sample is lost. However, measured rates agree well with data reported by Borths et al. [1].”

Borths et al. did a more thorough analysis of their data, and they calculated a corrected rate of 4.3 nmol/min/mg which is equivalent to 1/2 molecule per transport per minute, which is similar to our residence time of 40 seconds per substrate molecule.

5) In Figure 2 the dynamic fluctuations are interpreted as ATP hydrolysis events. Why do such events also occur before ATP is added? From visual inspection I cannot see more fluctuations in Fig. 2b/c than in Fig. 2a. Can the authors show a zoom-in of these fluctuations and quantify them? The number of events should be dependent on ATP

concentration. Such a study would strengthen the chapter dealing with the fluctuations as well as the interpretation of the data.

We think that we were not clear about the definition of dynamics or fluctuations, which appears to have caused a misunderstanding. The amount of fluctuations in single traces is quantified by grey bars in the right panel of Figure 2b and c. We clarified the definition in the text:

“We also observed that the fluorescence intensity of mainly acceptor fluorophore fluctuated much more after the addition of ATP than before – from the lower fluorescence level almost back to the initial level (Fig. 2b, middle and right panel). The dynamics of the fluctuations in our traces did not depend on laser intensity (data not shown) and our observed fluctuations are on a timescale that is much larger than what can be expected for blinking (milliseconds). Therefore we conclude that the fluctuations did not originate from blinking by the non-radiative triplet state of the fluorophore.”

Regarding the ATP dependence of these fluctuations, we show data for 0 ATP and for a (saturating) concentration of 2 mM ATP [Borths *et al.*]. We believe that additional time-consuming experiments with intermediate concentrations of ATP are not justifiable given the clear difference between the different ATP concentrations.

6) The study rests on the very low transport efficiency and high basal ATPase activity, which is not affected by vitamin B12. However, it would be key to provide direct data for the stoichiometry of uncoupling for the system used under identical conditions (see above). It is essential to know how many liposomes containing BtuCD-BtuF in the proper orientation are transport active and which are not. This is crucial to interpret the highly uncoupled ATP and transport turnover rates.

We performed a series of experiments to address this point, including ATPase assays and determination of the orientation of the proteins in the proteoliposomes (see also our response to point 4). In the discussion we added text that relates ATPase activity to transport activity, concluding that although we cannot observe both at the same time at the single-molecule level, we are looking at the same subset of transporters in both experiments.

“In the absence of tools to simultaneously measure ATPase activity and substrate binding in single-molecule studies, we base our reasoning on observed time scales of ATPase and transport rate (of which the ATPase rate is at least 10-fold higher) and occurrence numbers: the percentage of complexes that is responsive to ATP (18%, Fig. 1) is similar to liposomes that show export of substrate upon addition of ATP (11%, Fig. 4), indicating we are looking at the same subset of complexes.”

7) Based on the Venus-Fly trap mechanics, the opening and closing of substrate-binding proteins can be elegantly followed by smFRET. The reader may ask why the authors did not apply these established approaches to probe the opening/closing of Btu-F in relation to the ATP turnover as well as in the presence and absence of vitamin B12.

The reviewer raises an interesting proposal here. Indeed, there is published work addressing the conformational dynamics of the substrate binding proteins by smFRET, for instance of the glutamine/glutamic acid/asparagine transporter GlnPQ [Gouridis *et al.*, *Conformational dynamics in substrate-binding domains influences transport in the ABC importer GlnPQ*. Nat. Struct. Mol. Biol. 2015]. Here they show that this protein uses an induced fit mechanism. For these experiments to work, a binding protein is needed that shows substantial conformational changes upon substrate binding in order to visualize them by single-molecule

FRET (smFRET). However, for BtuF the distance changes between the two binding lobes are too small to be probed by smFRET (unpublished work from Gouridis). We updated the text in the introduction to give a better introduction to this substrate binding protein.

“A single substrate-binding protein BtuF completes the transporter. This SBP belongs to cluster A or class III and exhibits relatively small conformational changes upon substrate binding [13].”

8) It remains unclear why different sizes of liposomes were used for different topics. Please provide comments. In addition, it is important to experimentally confirm the diameter and the derivation by dynamic light scattering.

As a standard, we always use 100 nm liposomes. We want a high effective concentration of BtuF inside the liposome (work close or above the K_d), with just one molecule of BtuF. This can only be achieved by using 100 nm liposomes, in which 1 luminal molecule is equivalent to $\sim 3 \mu\text{M}$. In only one experiment in our manuscript we use liposomes of 200 nm. In this case we wanted to have the possibility for exchange of BtuF, so we needed multiple copies inside the liposome. As we did not want to change the effective concentration, this meant we had to use larger liposomes of 200 nm (by using even larger liposomes, we would risk multi-lamellar liposomes). We clarified this in the text and also included additional dynamic light scattering measurements to determine the diameter of our vesicles.

“In this experiment we used liposomes with a diameter of 200 nm to increase the number of BtuF molecules in the lumen (8 instead of 1) at the same concentration ($3.2 \mu\text{M}$) as the previous experiments that used liposomes with 100-nm diameter;”

“...and subsequently proteoliposomes were extruded at 100 nm (this diameter is always used, unless explicitly stated otherwise). Dynamic light scattering measurements on these samples show an outer diameter of 130 nm with a polydispersity of 15%; liposomes extruded at 200 nm measure a diameter of 160 nm with a polydispersity of 17%.”

9) It is yet ambiguous whether the intensity fluctuations really reflect on ATP hydrolysis. In order to support this idea, the authors need to analyze hydrolysis inactive mutants such as the E-to-Q substitution of the catalytic base. Those need to be included to strengthen the overall conclusions of the manuscript.

This point is raised by all reviewers and we realized that this is indeed a crucial point for interpreting our data correctly. We created an ATPase impaired mutant (BtuD E159Q), which we call BtuCD_{EQ}, and performed addition experiments. Throughout the entire manuscript we updated text and figures, of which the key changes are shown below:

- Figure 1b (ATPase rate) included the BtuCD_{EQ} mutant and confirms that there is no ATPase activity.
- Figure 1e is added: single-molecule experiments on BtuCD_{EQ} (see below). There are no quenching events upon and after addition of ATP. From this we can strongly conclude that the quenching events we do see in experiments with BtuCD_{cys} – both at the onset and after addition of ATP – are related to ATPase hydrolysis.

Specific points:

1) Important details of the experiments (ATP concentration, liposome, size etc.) should be included in each figure legend to simplify the understanding.

We now included these numbers in both figure and legend were appropriate.

2) Can the authors provide more details about the transporter labelling and the quantification of the labelling efficiency?

We added details to the methods section:

“While immobilized on the nickel column, BtuCD was labelled with the appropriate dye (Alex Fluor 488, Alex Fluor 555 or Alex Fluor 647, Thermo Fisher Scientific) at a concentration of 100 µg/ml in 50 mM KPi, 200 mM KCl, pH 7.5 (buffer A) + 0.03% DDM (buffer B) at 4 °C for 15 minutes while gently mixing. Reaction volumes were 500 µl + bed volume. ... Protein yields and labelling efficiencies were determined with SEC by measuring absorption at 280 nm and the appropriate dye wavelength.”

3) What is ultrapure water?

By ultrapure water we mean Type I pure water, also known as MilliQ. We updated the text.

4) Which objective was really used? Single-molecule expects would like to know these details. It is unclear why a 6-s filter was used. Please comment.

We used an Olympus ‘UApo N 100x O TIRF’ objective, as now can be found in the methods section.

The 6-s filter was used to filter background, which only contains noise on top of fluctuations on a slow timescale. The previous text could introduce some confusion on what trace the 6-s filter is applied, so we updated it:

“Integrated background counts, normalized to a circle with 7 pixel diameter and extracted from a nearby area devoid of peaks, were smoothed by a temporal median filter of 6 seconds and subtracted from the trace.”

5) Figure 1: The fluorophore is already present at BtuF before the incubation. This should therefore be included in the left panel of the illustration.

Assuming that the reviewer is referring to Fig. 1c, we explicitly did not include labels, because we are depicting a general setup here. Labels on BtuF are not present in all

experiments, so to avoid confusion we did not sketch them. We now explicitly state in the figure legend that “(fluorescent labels are omitted)”.

6) Legend of Fig. 1: it would read “marked with a yellow dot” instead of red dot.

This was an oversight and has been rectified.

7) Legend of Supplemental Figure 3: it should read “marked with a black line” instead of red line.

This was an oversight and has been rectified.

8) Supplemental Figure 9: The authors need to explain why the fluorescence does not stay constant after buffer exchange using their new setup of the flow-chamber (red curve). Which fluorescence molecules are detected?

We agree that the text was too brief in describing the method we used to create the graph and updated the figure legend:

“Both curves are measured in TIRF by introducing rhodamine B and measuring the total fluorescence intensity of non-specifically surface-bound dye molecules. With the typical flow speeds used in our experiments, 80% increase is reached in two seconds. Increasing the flow speed improves the sharpness of the transition, but also introduces focal drift in our current setup. With the new setup, a drop in intensity is observed after a few seconds, which could be caused by non-laminar flow effects.”

9) Supplemental Figure 7 shows strong fluctuations of another donor molecule with blinking events on the left. The same appears at the acceptor trace (bottom right). The authors need to comment on blinking and FRET events.

Although the reviewer marks this as a minor point, we believe that our text was confusing and could lead to major misunderstanding of the term fluctuations or dynamics. Therefore, we updated the main text to better describe how we define fluctuations and that these events are not due to blinking:

“We also observed that the fluorescence intensity of mainly acceptor fluorophore fluctuated much more after the addition of ATP than before – from the lower fluorescence level almost back to the initial level (Fig. 2b, middle and right panel). The dynamics of the fluctuations in our traces did not depend on laser intensity (data not shown) and our observed fluctuations are on a timescale that is much larger than what can be expected for blinking (milliseconds). Therefore we conclude that the fluctuations did not originate from blinking by the non-radiative triplet state of the fluorophore. The fluctuations could also not be explained by large distance changes between BtuCD and BtuF (visible as FRET) alone because most of the events were not characterized by anti-correlation of donor and acceptor intensities, but rather by changes in total intensity of both donor and acceptor combined (Supplementary figure 10).”

10) Supplemental Figure 1: Details of the transport conditions must be included into the legend.

The figure legend is updated:

“Uptake of [⁶⁷Co]vitamin B₁₂ in proteoliposomes. BtuCD and BtuF were purified and labelled as described before (see Methods), and reconstituted into proteoliposomes as described by Borths et al. [1]. Uptake of [⁶⁷Co]vitamin B₁₂ was measured essentially as described by Borths et al. [1], but with the omission of an ATP regenerating system. Liposomes were

loaded (see Methods) with 2 mM ATP and 10 mM MgCl₂, extruded at 400 nm and washed by means of centrifugation and resuspension. Reactions mixture contained ~3 mg/ml lipids and ~0.3 μM BtuCD and reactions were initiated by adding 1.0 μM BtuF and 0.1 μM of [⁶⁷Co]vitamin B₁₂ on the outside. Each measurement was performed in triplicate; mean and standard deviation are plotted. Measurements were performed in sets of two (labelled by same line colour and symbol shape) from the same batch of reconstitution. Uptake rates should not be considered as absolute numbers, because accumulated ADP might inhibit the transporter, there is a possibility of multi-lamellar 400-nm proteoliposomes and during the filtering step in the method sample is lost. However, measured rates agree well with data reported by Borths et al. [1].”

11) Figure 2 and 3: FRET traces should be shown. This would help the reader to differentiate between correlation and no correlation.

We added a new supplementary figure and made some changes in the main text:

Supplementary figure 10 – Complex formation observed with FRET – total intensity and FRET

Complementary to Fig. 2. For both (a) and (b), data in the left panels is equal to Fig. 2b and c. The right panels show total intensity (black, sum of donor and acceptor) and FRET (blue, acceptor divided by total intensity) traces.

“The fluctuations could also not be explained by large distance changes between BtuCD and BtuF (visible as FRET) alone because most of the events were not characterized by anti-correlation of donor and acceptor intensities, but rather by changes in total intensity of both donor and acceptor combined (Supplementary figure 10).”

12) Figure 2: what is the correlation between the donor/acceptor traces to the figure b, right part on the right with the sum up?

We updated the figure legend:

“The right panel shows the average of all traces where a drop in total intensity was observed upon introduction of ATP; the pair of traces shown in the middle panel is one of them.”

13) Figure 2: the significance of the fluctuations is unclear. ~20% offset from the initial signal and the fluctuations rise often to the initial values.

We addressed this point by our reply to point 9.

14) Figure 1: How many molecules are analyzed in each of these histograms? What is the explanation for the rise of the signal in Fig. 1c?

We added the total number of molecules analyzed to the figure legend.

It is unclear what the reviewer means by 'the rise of the signal in Fig. 1c'. In case he meant Fig. 1f (current numbering), the main text states:

*"The experiment resulting in the starting state of the above experiment was also performed: unlabelled BtuF was added to the outside of proteoliposomes not containing ATP. In 45% of the cases (corresponding to 82% of the complexes with right-side out orientation), an increase in fluorescence intensity was observed, corresponding with the binding of BtuF to those BtuCD complexes that were oriented right-side out (**Fig. 1c**, bottom panel, and **Fig. 1f**). From these experiments, we can conclude that a strong interaction between BtuCD and BtuF resulted in a high fluorescence intensity, whereas a change in interaction induced by ATP hydrolysis lowered the fluorescence. These findings are well supported by ensemble measurements (Supplementary Fig. 7)."*

Reviewer #2 (Remarks to the Author):

In this interesting manuscript, Goudsmits and coworkers report clever experiments at the single transporter level on the vitamin B12 importer BtuCD. This is very exciting work that provides novel insights into the working mechanism of a membrane transporter. The manuscript would benefit from further clarifications. The data also deserve better integration into what is currently known at the biochemical and cell biological level about BtuCD-F in relation to other ABC importers. A few key points are provided below.

1. In the summary, the BtuCD-F complex could be better introduced. The function of BtuF is not explained.

We recognize that the summary misses a short statement about the component of the complex, we therefore added the following sentence (we try to keep the summary as concise as possible):

“The periplasmic soluble binding protein BtuF binds the ligand; the transmembrane and ATPase domains BtuCD mediate translocation.”

2. In the Introduction the difference in structural and functional properties of the periplasmic binding protein deserve attention; in contrast to other binding proteins, BtuF might not show substrate binding via a venus flytrap mechanism, and this could have important implications for how substrate binding to BtuF is linked to interactions of liganded and unliganded BtuF to BtuCD. Another element that deserves further explanation is the differences in the degree of substrate stimulation of the ATPase activity between different binding protein-dependent ABC importers, and the position of BtuCD-F in this. The information should not only excite the expert but also inform the interested reader about the importance of what was found. The Introduction finishes with the question whether “ATP is merely required to reset the transporter”, but a clear answer regarding this point is missing at the end of the Discussion.

Also here we recognize that our transporter deserves better integration of what is currently known. We made major changes to the introduction, as can be found below:

“ABC transporters are membrane proteins that translocate substrates across a lipid bilayer [1]. They consist of two highly-conserved nucleotide binding domains (NBDs) that utilize energy of ATP binding and hydrolysis to drive conformational changes in the two transmembrane domains (TMDs), resulting in the formation of a pathway for substrate transport [2]. The TMDs are not conserved among all ABC transporters, and appear to have evolved from multiple ancestors [3]. Based on the different three dimensional architectures of the TMDs, ABC importers can be categorized as type I, type II, or energy-coupling factor (ECF) transporters [4-6], each with a different mechanism of transport [7-9]. Type I and II importers are complemented with soluble substrate-binding proteins (SBPs) that bind their ligands and deliver them to the TMDs. Based on structural properties, these SBPs can be further classified into clusters [10, 11].

The Escherichia coli (E. coli) vitamin B₁₂ transporter BtuCD-F is the best characterized type II importer. The homodimer BtuC spans the membrane and the two identical cytosolic ATPase domains BtuD form a sandwich dimer that couple chemical energy of 2 ATP molecules into structural changes of the full complex [12]. A single substrate-binding protein BtuF completes the transporter. This SBP belongs to cluster A or class III and exhibits relatively small conformational changes upon substrate binding [13]. Extensive structural and biochemical characterization of BtuCD alone or in complex with BtuF has provided the framework for understanding the mechanism of vitamin B₁₂ transport: crystal structures have

revealed several intermediate states in the transport cycle [14-16], the gating mechanism of the TMDs upon ATP hydrolysis has been investigated with EPR techniques [17, 18], and biochemical characterization of the complex in detergent and proteoliposomes has given insight into the molecular steps underlying transport [19, 20]. These studies all suggest that BtuCD-F employs a different mechanism than that described by the alternating-access model for type I importers. Nonetheless, many questions remain unanswered with different studies in detergent and lipid environments showing varying results. Does the soluble substrate-binding protein BtuF remain bound to the BtuCD transmembrane complex during the cycle [19], or does ATP hydrolysis release the binding protein into the periplasmic space [20]? Is the substrate immediately transported, and is ATP merely required to reset the transporter [20]? Why is the ATPase activity at least one order of magnitude higher compared to transport of vitamin B₁₂ – in other words, why is there no strong coupling between ATP hydrolysis and substrate translocation such as observed for the well-studied type I maltose importer MalFGK [21]? In order to address these questions, we have employed single-molecule fluorescence techniques to follow individual BtuCD-F proteins reconstituted in liposomes through time and to directly observe steps of the transport cycle. Ultimately, we interpret our results in the light of different existing models for transport by BtuCD-F.”

As to answering the question “...ATP is merely required to reset the transporter”, we now included this explicitly in the discussion. Also see a more advanced explanation at point 9.

“ATP binding and hydrolysis is required to continuously reset the transporter, i.e. induce structural changes that allow for substrate binding on the periplasmic site.”

3. The text frequently refers to consistency between the current data and “ensemble measurements”, “previous bulk measurements”, “previous ensemble characterizations” etc., but the details of what is exactly similar, is left to be discovered by the reader; it is the authors who should explain and emphasize this in sufficient detail.

We agree that it is better to explain directly what is exactly measured and/or similar in referring to previous experiments. Occurrence of the “previous bulk/ensemble measurements” have been replaced or the context has changed:

- *“This fraction of apparent active transporters corresponds well with the previously measured functionally competent fraction in bulk measurements [19].”*
- *“Our findings are strongly supported by previous ensemble characterizations of the transporter embedded in liposomes where the authors looked at exchange of labelled versus unlabelled BtuF and no exchange is observed [16].”*

Results

4. p. 5: “While the fluorescence was high in the absence of nucleotide, in a fraction of liposomes (8%) it decreased rapidly upon addition of ATP to the solution surrounding the liposomes.” The author subsequently state that this represents the fraction of active transporters. What is meant with "active" here? Given the known ATP concentration and known binding constant of BtuCD for ATP is should be possible to calculate the fractional occupation of ATP binding sites by ATP. Given the experimental conditions, this number will most likely be greater than 8%. What happens with the other 92%?

The reviewer notes the fraction of 8% that is responsive to ATP, and asks whether this is because of fractional occupation of ATP. In all experiments, we use saturating concentrations of 2 mM ATP (Borths et al.), so all transport complexes should respond.

However, many do not, because they are inactive – defined as not responsive to ATP. Now, we also take into account the orientation of complexes in the liposome, and conclude that 18% is active. The possibility to separate the inactive and active fraction, shows the power of single-molecule experiments. Importantly, the fraction of active complexes in our single molecule experiments matches very well with the functionally competent fraction reported by Borth *et al.* We updated the text to clarify our point:

“While the fluorescence was high in the absence of nucleotide, in a fraction of liposomes (8% of the total) it decreased rapidly upon addition of ATP to the solution surrounding the liposomes. Taking into account that 45% of the transporters are oriented inside out (Supplementary Fig. 2), i.e. with the BtuD ATPase domains facing to the outside, only 18% of the complexes respond to ATP – or are active. This fraction of apparent active transporters corresponds well with the previously measured functionally competent fraction in bulk measurements [19].”

5. p 7: “The fluctuations could not be explained by FRET alone because many of the events were not characterized by anti-correlation of donor and acceptor intensities.” Interesting statement but not clear.

We made textual changes and added a new Supplementary Figure because the statement is indeed not clear. We meant that fluctuations are not anti-correlated in donor and acceptor, and they are still present when summing the intensities.

“Therefore we conclude that the fluctuations did not originate from blinking by the non-radiative triplet state of the fluorophore. The fluctuations could also not be explained by large distance changes between BtuCD and BtuF (visible as FRET) alone because most of the events were not characterized by anti-correlation of donor and acceptor intensities, but rather by changes in total intensity of both donor and acceptor combined (Supplementary figure 10).”

Supplementary figure 10 – Complex formation observed with FRET – total intensity and FRET

Complementary to **Fig. 2**. For both (a) and (b), data in the left panels is equal to **Fig. 2b and c**. The right panels show total intensity (black, sum of donor and acceptor) and FRET (blue, acceptor divided by total intensity) traces.

6. p. 9: “After summing all traces that show a decrease in fluorescence intensity at zero seconds, we extracted an average residence time of 40 +/- 10 seconds per substrate molecule in the transporter”. This is a key conclusion in the paper, but this statement and explanation are really too short.

We agree with the reviewer that is a key conclusion in our manuscript and therefore this statement deserves much better explanation. Although details of the ‘we extracted an average residence time of 40 +/- 10 seconds’ can be found in the methods section, we elaborated on this statement in the main text:

“When the residence time of a single vitamin B₁₂ molecule was described by an exponentially decreasing distribution, the distribution of total residence times for multiple molecules is given by a rise-and-decay function (see Methods). Since the number of substrate molecules inside a liposome follows a Poisson distribution, the total residence time of all vitamin molecules in a single transporter in one proteoliposome is described by a convolution of both aforementioned distributions. Summing all traces that show a decrease in fluorescence intensity upon addition of ATP, directly yields the cumulative convoluted distribution. We fitted this distribution and extracted an average residence time of 40 +/- 10 seconds per substrate molecule in the transporter (Fig. 4b, right panel; Methods).”

Discussion.

7. The authors state: “Although it is currently not possible to simultaneously measure ATPase activity and substrate presence in single-molecule studies” [.....]. However, as the main conclusion of this paper is based on a comparison of the residence time for vitamin B12 binding versus rate of ATP hydrolysis, it is vital to measure in the current test system with current experimental conditions what the rate of ATP hydrolysis is, even if this would be measured non-simultaneously and in bulk.

The reviewer addresses a very important point here that is also addressed by the other reviewers. We included the measurements. Moreover, we included extra experiments to determine the orientation of the BtuCD protein in the liposomes to even better interpret rates.

We added a new Supplementary Figure (2) which reveals the orientation. We found that 55% is oriented right-side out; 45% is oriented inside out.

We added a bar chart with ATPase rate of various mutants we use (Fig. 1b). These numbers match the timescale at which we see ATP-dependent fluctuations in single-molecule traces.

“The dynamics that we attribute to conformational changes induced by ATP turnover took place on a timescale of seconds, which matches data obtained by ATPase rate experiments as described above (Fig. 1b: 15 ATP molecules per BtuCD-F complex per minute, not corrected for orientation and stoichiometry of 2 ATP molecules per BtuD sandwich dimer).”

We also changed the text in the discussion paragraph to better reflect the numbers and timescales we measure:

“In the absence of tools to simultaneously measure ATPase activity and substrate binding in single-molecule studies, we base our reasoning on observed time scales of ATPase and transport rate (of which the ATPase rate is at least 10-fold higher) and occurrence numbers: the percentage of complexes that is responsive to ATP (18%, Fig. 1) is similar to liposomes that show export of substrate upon addition of ATP (11%, Fig. 4), indicating we are looking at the same subset of complexes.”

8. What is the role of the Btu-F binding protein in transport? Would it be possible for the ligand to directly bind to and dissociate from BtuCD without involvement of BtuF,

and could this explain the high rate of ATP hydrolysis compared to the rate of transmembrane transport of the substrate?

We will discuss this point together with 9.

9. What is the role of ATP binding and hydrolysis in transport by BtuCD-F? Does ATP hydrolysis occur in response to ligand binding to BtuCD? If ATP would merely reset the transporter to complete the transport cycle, then what drives the translocation process? Is transport of B12 directed down a concentration gradient into cells that utilize this ligand in metabolism, or uphill in a reaction that would require input of metabolic energy? Some discussion would be useful here.

The reviewer asks for more discussion about the role BtuF and ATP binding and hydrolysis in the transport model. We expanded the discussion substantially to answer all questions in point 8 and 9, and also included a discussion that puts our transporter in perspective to other transport systems.

“In our model, we directly show continuous turnover of ATP – with or without substrate. ATP binding and hydrolysis is required to continuously reset the transporter, i.e. induce structural changes that allow for substrate binding on the periplasmic site. Some of this energy might also induce structural changes that create a pathway for substrate escape to the cytosol, however, currently the structural evidence for that change is missing. As vitamin B₁₂ can be translocated against a concentration gradient, input of metabolic energy is required though [19]. An additional role of BtuF – besides capturing substrate with high affinity – is probably to close the periplasmic site of the transporter when a translocation channel is formed to the cytoplasm. Dissociation of BtuF from BtuCD is rare, and not an obligatory step for transport. Occasional dissociation may be related to escape from off-path conformations such as the asymmetric state with collapsed cavity trapped in a crystal structure [27]. The futile energy consumption in the absence and even presence of substrate, also previously noted in the ABC exporter P-glycoprotein [28] and other ABC importers [9], might raise questions about the biological purpose. We speculate that this is a mechanism to effectively capture and transport rare essential substrates such as some B vitamins. Type I ABC importers for highly abundant nutrients, such as the maltose transporter MalFGK, seem to show tight coupling between ATP hydrolysis and substrate transport [21]. Most likely there is also no evolutionary drive to optimize transporters for rare substrates as total ATP consumption for these transporters is negligible compared to total cellular ATP consumption.”

10. “In our model, dissociation of BtuF from BtuCD is rare, and not an obligatory step for transport.” How is this interaction structurally maintained; is there any information available from crystal structures regarding this point?

The interaction between BtuF and BtuCD is structurally only addressed in the fully bound state – i.e. both lobes of BtuF bind the parts of TM5 in BtuC [Hvorup *et al.*, 2007]. There is no crystal structure available regarding a partially bound (i.e. with one lobe) BtuF that allows for substrate entrance. It is questionable whether this can be found in a crystal, as flexible elements are hard to resolve. The publication by Hvorup *et al.* might already indicate for this, we cite: “...whereas in the region of BtuF, the electron density was not as good (fig. S1A). This probably reflects the flexibility of BtuF when bound to the transporter or the absence of lattice contacts involving BtuF.”

Reviewer #3 (Remarks to the Author):

Goudsmits et al. performed single-molecule imaging experiments to gain a deeper understanding of the transport mechanism in the VitB12 importer BtuCD-F, a type II class ABC transporter. BtuCD-F from *E. coli* is a well-studied model system and has been extensively characterized both structurally and biochemically. However, the authors indicate that a number of open questions remain which pertain to the order of events in the transport cycle and whether VitB12 transport is tightly coupled to ATP hydrolysis that must be answered in order to achieve an understanding of the transport (import) mechanism. These biological questions are certainly interesting and a single-molecule approach seems like an appropriate application to probe the types of questions that the authors would like to address.

After carefully reviewing the written manuscript, which is concisely written and appears to be appropriately referenced, my sense is that there are considerable shortcomings in the execution and interpretation of the experiments that currently preclude this manuscript from publication. Aside from the technical issues discussed below, one of the main conclusions presented, namely that BtuCDF non-productively hydrolyzes ATP even in the absence of B12, is somewhat troubling. While such futile energy consumption has been previously noted, the biological purpose for such a mechanism is entirely unclear?

In our experiments, both single-molecule (Fig. 1/2) and ensemble (Fig. 1b, newly included), we showed ATPase activity of the complex in absence of substrate. Also earlier studies on the same transport complex (Borths *et al.*, 2005), on other ECF-type ABC importers (Swier *et al.*, 2016) and on the ABC exporter P-glycoprotein (Shapiro *et al.*, 1995) showed uncoupling of ATPase activity and transport.

The biological relevance of such mechanism is based on speculation, but we tried to include a possible explanation in our discussion:

“The futile energy consumption in the absence and even presence of substrate, also previously noted in the ABC exporter P-glycoprotein [28] and other ABC importers [9], might raise questions about the biological purpose. We speculate that this is a mechanism to effectively capture and transport rare essential substrates such as some B vitamins. Type I ABC importers for highly abundant nutrients, such as the maltose transporter MalFGK, seem to show tight coupling between ATP hydrolysis and substrate transport [21]. Most likely there is also no evolutionary drive to optimize transporters for rare substrates as total ATP consumption for these transporters is negligible compared to total cellular ATP consumption.”

Perhaps most importantly, the authors state at the onset that they are working with a preparation of protein that has a remarkably low active fraction: as stated on page 5, only 8% of the protein is active.

We state that 8% of the analysed liposomes is active – i.e. responsive to ATP. This number is uncorrected for orientation of the complex. We performed experiments to determine the orientation of BtuCD in our liposomes, and found that 55% is oriented right-side out; 45% is oriented inside out (Supplementary Fig. 2). After correction for this fraction, we conclude that 18% is active. This number is similar to previously reported bulk experiments on the active fraction of reconstituted BtuCD-F by Borth *et al.*, reinforcing the correspondence between our single molecule experiments and bulk experiment.

We update the text to include these numbers:

“While the fluorescence was high in the absence of nucleotide, in a fraction of liposomes (8% of the total) it decreased rapidly upon addition of ATP to the solution surrounding the liposomes. Taking into account that 45% of the transporters are oriented inside out

(Supplementary Fig. 2), i.e. with the BtuD ATPase domains facing to the outside, only 18% of the complexes respond to ATP – or are active. This fraction of apparent active transporters corresponds well with the previously measured functionally competent fraction in bulk measurements [19].”

While such low activity may be a good argument for performing single-molecule investigations (heterogeneous ensemble)...

The possibility to separate the inactive and active fraction, does indeed clearly show the power of single-molecule experiments. This inactive fraction is also present in ensemble experiments, in which one can only correct for it afterwards (Borths *et al.*, 2005); an intrinsic exclusion of this fraction is only possible with single-molecule experiments.

... this limitation seems to significantly compromise all downstream experiments as the statistics resulting from the relatively low number of molecules per imaging experiment appear lacking. In some instances, it would appear that at least some fraction of the conclusions may have been drawn from just a few carefully-chosen traces (for example, single traces in Fig. 2 are used to support stable BtuF binding in the absence of ATP, and quenching with FRET in the presence of ATP).

Our conclusions were not based on a handful of traces. After analyzing over 1000 fluorescence traces, our algorithm found 50 traces that show FRET and have a response to ATP (Fig. 2b, right panel). The donor and acceptor traces in the middle panel are representative for ‘active’ transporters.

These considerations, and the absence of complementary bulk activity assays, aside from showing that the protein has some level of transport function, and/or ensemble fluorescence measurements of the specific activities described (such as ligand binding studies), make it extremely difficult to evaluate the robustness of the conclusions drawn from individual experiments.

The reviewer addresses a very important point here that is also raised by the other reviewers. We included the measurements of ATPase rates of various mutants we used (Fig. 1b). These numbers match the timescale at which we see ATP-dependent fluctuations in single-molecule traces.

“The dynamics that we attribute to conformational changes induced by ATP turnover took place on a timescale of seconds, which matches data obtained by ATPase rate experiments as described above (Fig. 1b: 15 ATP molecules per BtuCD-F complex per minute, not corrected for orientation and stoichiometry of 2 ATP molecules per BtuD sandwich dimer).”

As for ligand binding studies, we will address this while discussing figure 4 below.

An additional complicating feature of the experiments described is shown in Figure 1, which shows that the “activities” defined in the present study are based on relatively small changes...

We do not agree and argue that a decrease in fluorescence at t=0 of roughly 50% is not “small” (Fig. 1d).

... in fluorescence intensity for which there is no physical basis for interpretation. Are the photophysical properties of Alexa 555 well characterized? Why not use a donor fluorophore for which there is literature regarding the basis of fluorescence changes (ie. Cy3).

Quenching is based on the change of the local environment of the dye. Other dyes, such as Cy3, have been used, and they show the same quenching behaviour. We now explicitly state this in the text. With substrate-addition experiments (absence vs. presence of ATP and/or BtuF, Fig. 1f, Supplementary Fig. 5 and 7) we deduce how BtuF is involved in the fluorescence quenching:

“The quenching of the fluorescence suggests an alteration of local protein environment of the dye and thus a conformational change of the complex (Fig. 1d, middle) [24]. Other dyes such as Cy3 and tetramethylrhodamine (TMR) manifest the same quenching behaviour (data not shown). The distribution of quenching events plotted in a histogram (Fig. 1d, right) revealed an ATP response time in our flow cell of 1.9 +/- 0.5 sec (see Methods). When we performed the same experiment in the absence of BtuF, no events were observed (Supplementary Fig. 5), indicating that the observed decrease in intensity was mediated by interaction with BtuF.”

What causes fluorescence to decrease in Alexa555 in the absence of vitamin B12 (Fig. 1c); what causes fluorescence to increase in the absence of B12 (Fig. 1e)?

As stated above, fluorescence decrease is caused by a change of local environment of the dye. The increase corresponds to binding of BtuF (and thus also a change of local environment of the dye). We clarified the text:

“In 45% of the cases (corresponding to 82% of the complexes with right-side out orientation), an increase in fluorescence intensity was observed, corresponding with the binding of BtuF to those BtuCD complexes that were oriented right-side out (Fig. 1c, bottom panel, and Fig. 1f).”

Can the authors show that one molecule can exhibit both effects in a biologically regulated manner?

Unfortunately we are currently unable to perform this experiment, as currently available technical capabilities do not allow us to. (In the experiments we performed, we probed transporters in different orientations as depicted by Fig. 1).

Are the observed changes in fluorescence intensity mitigated by mutations in the protein that block transport activity?

This point is raised by all reviewers and we realized that this is indeed a crucial point for interpreting our data correctly. We created an ATPase impaired mutant (BtuD E159Q), which we call BtuCD_{EQ}, and performed addition experiments. Throughout the entire manuscript we updated text and figures, of which the key changes are shown below:

- Figure 1b (ATPase rate) included the BtuCD_{EQ} mutant and confirms that there is no ATPase activity.
- Figure 1e is added: single-molecule experiments on BtuCD_{EQ} (see below). We clearly see no quenching events occur upon and after addition of ATP. From this we can strongly conclude that the quenching events we do see in experiments with BtuCD_{cys} – both at the onset and after addition of ATP – are related to ATPase hydrolysis.

In summary, these shortcomings of the data severely limits the impact of the work and appears to call into question the reliability of the conclusions.

We believe we have addressed the major concerns of the reviewer, including new experimental data that reinforces our conclusions. Below we will discuss some specific comments in more detail.

Specific comments:

Figure 1: Interaction between BtuCD and BtuF in absence of B12 substrate, in the presence of ATP: The quenching of single, Alexa 555 on reconstituted BtuCD was shown to occur upon addition of ATP outside the proteoliposomes, and was dependent on the presence of BtuF, and apparently occurs only upon ATP hydrolysis as quenching does not occur upon the addition of AMP-PNP. It is not explained why the specific labeling sites were picked, in particular for BtuCD where the quenching of fluorescence is argued to be due to a conformational change.

The labelling site on BtuF is chosen on the middle of the backbone connecting the two lobes, such that this position is insensitive to the orientation. The labelling site of BtuC is chosen on a periplasmic loop connecting two transmembrane helices, that is not strongly affected by the asymmetry of BtuCD [Hvorup *et al.*, 2007]:

“To visualize BtuCD and BtuF, we created single cysteine mutants of BtuC (Q111C, on the periplasmic loop connecting transmembrane helix (TM) 3 and 4) and of BtuF (D141C, pointing outward in the middle of the alpha helix connecting the two lobes) that allow for specific coupling of fluorescent labels to each of these proteins (Fig. 1a).”

We tried several different mutants of BtuCD, but they did not express well or were inactive after labelling.

It is difficult to determine where the quenching step occurs in the sample traces shown as there are numerous fluctuations in fluorescence intensity.

The number of single molecules (N) analyzed in the histograms on the right are not shown and the methods used to generate these histograms are not clearly described and controls are not delineated. Were the traces analyzed by eye or was there some idealization of the data?

The quenching step is determined by our algorithm (see Methods) as the first moment where the intensity of the trace falls down 3.5 times the standard deviation of the initial signal. We added the number of molecules analyzed to the figure legend. As described in the main text,

negative (controls) and positive experiments only differ by the reaction buffer that is used; the experimental procedures and analysis of the data are exactly the same.

Figure 2: FRET between BtuCD and BtuF: High FRET is observed when BtuCD-F complex formed: In the absence of B12 you see a drop in acceptor fluorescence upon addition of ATP, but also a drop in donor fluorescence, apparently due to quenching as in the first experiment.

It is difficult to make strong conclusions about this experiment in general given the number of confounding variables. Qualitatively it is difficult to tell the differences or similarities between the single traces in 2b and 2c, but there seems to be more dynamics in donor fluorescence in presence of B12.

Sample traces in Fig. 2b and 2c are two of the traces that are used to build statistics in the right panels of these figures. The observed difference in dynamics between donor and acceptor traces can be due to heterogeneity of the sample; the general behaviour – from which we draw conclusions – is summarized in the right panels. We clarified the text.

What are the fluctuations in Alexa 647 intensity?

We did not observe any intrinsic fluctuations in Alexa 647; data is not shown.

Figure 3: Stability of BtuCD-F complex: If there is an exchange between two different BtuF molecules that are bound to BtuCD, there would be an observed FRET change since there are two BtuF populations in the liposomes, each labeled with a different dye.

No change in FRET can be discerned in the two traces shown. Some sort of ensemble statistics is absolutely necessary. Were any complexes observed on the ‘blue’ Alexa 488 channel? If not, why?

The experiment described here can only be done without a positive control – as far as we know there is no possibility to induce an exchange of SBPs as verification. Therefore, we agree that these data may seem unsatisfactory to the reviewer. Showing statistics for something that does not happen and without a control is unavailing, so we did not include these.

We do want to emphasise that these results have to be interpreted in the context of other data that we show, especially Supplementary Fig. 9 (current numbering). Based on the latter results we see that exchange is rare and we inferred a timescale of several minutes.

With the current setup, we were unable to visualize proteins simultaneously in all three channels. However, we were able to see complex formation in the ‘blue’ and ‘red’ channel simultaneously, but this required an elaborate change of filters, and data is not shown.

Figure 4: B12 Transport Assay.

Clear quenching of fluorescence is observed for BtuF alone in presence of B12 (Fig. 4a). As this appears to be the ‘loudest’ signal in the manuscript, additional analyses of this effect would seem reasonable such as a binding curve or waiting time distribution, to show its concentration dependence.

The quenching signal of vitamin B12 on BtuF could indeed be a very nice probe to measure binding kinetics of substrate to the SBP. Indeed, this has recently been performed by Mireku and co-workers [Mireku *et al.*, *Conformational Change of a Tryptophan Residue in BtuF Facilitates Binding and Transport of Cobinamide by the Vitamin B12 Transporter BtuCD-F*. Sci. Rep. 2017].

Given that B12 has a 15 nM affinity for BtuF (page 9) does this mean that all experiments in the paper containing B12 have this quenched fluorescence baseline?

The quenching effect is only present when using Alexa Fluor 488 (not with any of the other fluorophores), as explained in the Supplementary Table 1. This is also reflected by data in Figure 2: here the amount of fluorescence quenching is independent of vitamin B12 concentration. We changed the text:

“This substrate quenching effect is clearly different from the quenching effect observed in Fig. 1 and 2 because the former was observed only with Alexa Fluor 488-labeled BtuF, and did not occur with Alexa Fluor 555 and 647 (Supplementary Table 1), whereas the latter effect was insensitive to substrate.”

Given this is nearly an 80% change in fluorescence intensity, it is difficult to understand the relative scale of the transport data. Even a single B12 molecule would quench the sensor (Kd: 15 nM, single molecule of B12 in lipo ~ 3 μM), and without an experiment where no B12 is added to the inside of the liposomes to compare with ...

The quenching of vitamin B12 in our complexed transporter is different than the 75% we see with BtuF only (Fig. 4). We explained this in the main text:

“Our results also suggest that the substrate resides in a different place than the BtuF binding pocket as the quenching effect in the full transporter is lower than in BtuF alone (Fig. 4a and b), although we cannot exclude that a photophysical effect reduced the fluorophore quenching in the full complex.”

The experiment without B12 loaded was already performed and presented in Supplementary Figure 11 (current numbering). Here we see that no events occurred (above background) where the fluorescence quenches upon addition of ATP. All traces remained flat. Now we included statistics about ATPase and transport active complexes:

“In the absence of tools to simultaneously measure ATPase activity and substrate binding in single-molecule studies, we base our reasoning on observed time scales of ATPase and transport rate (of which the ATPase rate is at least 10-fold higher) and occurrence numbers: the percentage of complexes that is responsive to ATP (18%, Fig. 1) is similar to liposomes that show export of substrate upon addition of ATP (11%, Fig. 4), indicating we are looking at the same subset of complexes.”

... it would appear that depletion of internal B12 is not complete by 200 seconds (Fig. 4b. right). Why is this the case?

Based on the distribution of vitamin B12 molecules in liposomes (see Methods for details), there is a fraction of liposomes that has more than the average number of 3 substrate molecules and the depletion will not be complete after 200 seconds.

Shouldn't the fluorescence intensity ultimately achieve an 80% higher intensity?

We explained above the possible causes of the percentage of less than 80%. Importantly, in the traces shown in Fig. 4b the intensity restored to the initial value.

How does this signal change when mutations are made in the active site that block B12 binding or transport? Such controls seem essential to verify the nature of the signal being measured.

To our knowledge there is currently no information available in literature that describes a mutant that blocks binding or transport of substrate, but is otherwise still active and binding of BtuF to BtuCD is not affected. Screening for such mutant is not feasible within the context of the revision.

Moreover, we question what the relevance is of a transport/binding-disabled mutant where the signal does not change over time, as this will not allow us to differentiate between active or inactive transporters in our single molecule measurements.

In Sup.Fig.7, a single trace shows periods of loss of acceptor fluorescence. The authors interpret this as unlabeled BtuF substituting for labeled BtuF. This behavior is informative if true, but could also be explained by photoblinking. No photoblinking and clear evidence must be provided to exclude such an interpretation. Alternating laser excitation may be of use here.

We did experiments with different laser power to exclude photoblinking. We clarified this in the main text:

“The dynamics of the fluctuations in our traces did not depend on laser intensity (data not shown) and our observed fluctuations are on a timescale that is much larger than what can be expected for blinking (milliseconds). Therefore we conclude that the fluctuations did not originate from blinking by the non-radiative triplet state of the fluorophore.”

(The second trace in this figure is clearly and admittedly aberrant and should not have been shown.)

It is unclear to us what the reviewer means with this statement.

Reviewers' comments:

Reviewer #1 (Remarks to the Author):

The authors tried to cover most of the concerns raised by the reviewers in their revised manuscript. Overall the quality and conclusion have been significantly improved. However, one major area of critics remains unaddressed which is essential. The work reports mainly on conformational changes detected by fluorescence quenching, which are interpreted as single ATP hydrolysis events. A convincing correlation of these events at different ATP concentrations has not been performed. Based on the indirect observation of single turnover events, the correlation with the ATPase activity at various ATP concentrations is essential.

Importantly, the revised version now defines that only 18% of the BtuCD-F preparation is ATPase active. The manuscript would benefit from experiments demonstrating that these ATPase active complexes are indeed able to transport vitamin B12. Again, this is critical for the overall conclusions. The statement that the same subset of active complexes is observed by ATPase and transport is far-fetched.

Along the same line, the title is still misleading to the general readership as no conclusions on single translocation events or a productive transport cycle can be made. Based on fairly ill-defined reconstitution, lipid environment or transport mechanism, BtuCD-F requires 30 or even much more ATP hydrolysis to occasionally move one vitamin B12 molecule across the membrane, depending with lab performed the assays. Therefore, the study does not report on the productive transport mechanism but rather on futile cycles of BtuCD-F. The title should read 'single-molecule visualization of ATP hydrolysis events within the vitamin B12 ABC importer BtuCD-F'.

Reviewer #2 (Remarks to the Author):

I thank the authors for their useful responses and revisions in the text and presentation of data. I have two important remaining questions, which are included below.

1. "Under conditions in which active transport takes place, a vitamin B12 molecule remains bound to the protein complex for tens of seconds, during which several ATP hydrolysis cycles can take place, before it is being transported across the membrane." As the ATPase measurements are done with a population of transporters of which 18% is transport-active (line 118), how do the authors know that vitamin B12 binding and multiple rounds of ATPase are directly linked and occur at the same transport complex? In other words, could the reconstituted proteins that are probed in ATPase measurements contain different sub-populations, and could non-transport-active complexes still exhibit ATPase activity? Given the published data on ABC transporters, for which even isolated NBDs exhibit ATPase activity, this would not be an unreasonable assumption. Along similar lines: how do the authors know in line 186 "15 ATP molecules [hydrolyzed] per BtuCD-F complex" that it is the full complex that gives rise to ATP hydrolysis in the experiments? As this could be perceived as a weak point in the paper, it will be important for the authors to discuss this further in the main text.

2. Could the suggestion of uncoupled transport (line 282-283) also be explained if BtuCD-F would transport alternative substrates in the test system, in addition to vitamin B12? The authors refer to P-glycoprotein (line 320-321) for which transport of lipids and detergents in the experimental system has been raised as an explanation for the basal ATPase activity in the absence of drugs. Is this type of explanation relevant for your research data? This point requires discussion in the manuscript.

Reviewer #3 (Remarks to the Author):

The experimental approaches in this manuscript represent interesting and potentially fruitful lines of investigation that enable the functions of the BtuCD-F system to be quantified at the single-molecule scale. Yet, the nature of the signals obtained, which are based principally on fluorescence intensity changes, severely limit the analyses presented and largely reduce the conclusions to qualitative correlations.

While ATP-turnover assays have been added, and the authors have made a significant efforts to address the majority of the concerns put forward during initial review, the main conclusions of the manuscript seem to indicate what has been put forward previously: that BtuCD futilely hydrolyzes ATP in the absence of substrate and that BtuF binds stably to BtuCD (the idea that it is a breakthrough that this occurs in a lipid bilayer and not just detergent micelles seems questionable). Although the authors have commented on the observation (and critique) that the BtuCD/F system burns ATP non-productively to suggest that this may be advantageous to vitamin scavenging. Such a model would appear to be an outlier with respect to what has been shown for other biological systems, where energy conservation and energy efficiency are put forward as paramount to evolutionary fitness. Supporting this view, small molecules that uncouple energy expenditure from function in living systems are typically antibiotics/antiproliferative agents. The other key conclusion presented, which is based in part on what the authors refer to as a three color experiment (Fig.3) is that BtuF stays bound to BtuCD while ATP hydrolysis occurs. This conclusion is inferred from their measurements indicating that ATP turnover occurs relatively quickly (on the order of seconds time scale fluorescence fluctuations) and the multicolor imaging studies discussed surrounding Fig.3, which have been interpreted as providing the conclusive evidence that BtuF dissociation is slow (on the order of minutes). As it stands Fig.3 is unconvincing, however: there are only two traces shown, neither of which shows a photobleaching event to confirm that FRET was indeed being observed; and if the conclusion is that 488-555 traces and 555-647 traces show highly stable complexes and no evidence of exchange, why not show one of each kind of trace to substantiate claim that this is a three-color experiment? An alternative, bulk method like liposome pelleting or pull down would seem necessary to provide supporting evidence for the single-molecule interpretation that the complex is highly stable.

While there is little doubt that the efforts presented are substantial and unique, these questions about the nature and novelty of the conclusions drawn brings up one of the foundational concerns I have about this work: why is only 8% (or 18% now that sidedness has been taken into consideration) of the protein included in the authors' analysis. While it is appreciated that others have reported similar difficulties working with this system, have previous structural and functional analyses of BtuCD(F) (i.e. like the ATPase assay in Figure 1) only focused on such a small subpopulation of active transporters?

While the authors focus on this subpopulation because it is the only fraction that provides the futile ATP hydrolysis signal, and I cannot argue that the conclusions drawn from this subpopulation are what they are, I am left with the sense that the present body of work would benefit from functional analyses of majority of molecules that are currently discarded. For instance, how do the single-molecule ATP hydrolysis assays compare to the bulk studies: does the majority of protein hydrolyze ATP or only the small subpopulation? Do the majority of proteins transport (ie. do the authors sees the rise if fluorescence show in Fig.4b for all proteins or for just an 8% subpopulation)? Is it possible that the conclusions that would be drawn about the majority of protein would be different than the 8%, for instance, does this fraction have a tighter coupling between ATP binding/hydrolysis and transport and ATP hydrolysis events are just not observed because the signal-averaging procedures employed mask the single (or few) fluorescence fluctuations that are present?

In summary, although the revised manuscript is indeed substantially improved by the new

experiments and discussion that have been included, in my view the aforementioned issues, together with the general considerations that there is little to effort made to show the reproducibility or statistical significance of the findings presented and profoundly new and obvious biological insights are lacking, stipulate that further refinements to the manuscript are needed prior to publication.

Reviewer #1 (Remarks to the Author):

The authors tried to cover most of the concerns raised by the reviewers in their revised manuscript. Overall the quality and conclusion have been significantly improved. However, one major area of critics remains unaddressed which is essential. The work reports mainly on conformational changes detected by fluorescence quenching, which are interpreted as single ATP hydrolysis events. A convincing correlation of these events at different ATP concentrations has not been performed. Based on the indirect observation of single turnover events, the correlation with the ATPase activity at various ATP concentrations is essential.

In our manuscript we present two independent experimental strategies to show that conformational changes detected by fluorescence quenching, are caused by ATP hydrolysis events. First, we show that the fluctuations do not occur when the non-hydrolyzable ATP analogue AMP-PNP is used instead of ATP (Supplementary Fig. 5, original manuscript). Second, we show that in the presence of ATP the fluorescence fluctuations are absent when a mutant (E159Q), which is unable to hydrolyze ATP, was used. These two independent lines of experimental evidence are strong. We do not understand how the repetition of the experiments at different ATP concentrations would make this conclusion any stronger.

The experiments that the reviewer requests represent a massive amount of work, and without conviction that they would contribute anything to the conclusion that the fluorescence fluctuations are caused by ATP hydrolysis, we cannot justify this effort.

Importantly, the revised version now defines that only 18% of the ButCD-F preparation is ATPase active. The manuscript would benefit from experiments demonstrating that these ATPase active complexes are indeed able to transport vitamin B12. Again, this is critical for the overall conclusions. The statement that the same subset of active complexes is observed by ATPase and transport is far-fetched.

This point is raised by all three reviewers, and therefore the response below is identical for all of them.

Formulated in slightly different ways all reviewers ask the question whether the 18% of complexes showing ATPase activity (as measured by fluorescence fluctuations, Fig. 1 and 2) corresponds to the same fraction of complexes (11%) that transports B12 as probed by the BtuF-AF488 (de)quenching assay (Fig. 4). Admittedly, in the absence of tools to simultaneously measure ATPase activity and substrate binding in single-molecule studies, the relation between ATPase activity and B12 transport that we present is indirect: we interpret an initial quenching of BtuF-AF488 fluorescence upon ATP addition as proxy for ATPase active complexes as defined by Fig. 1.

Even though we find this assumption reasonable, we do agree further confirmation would be desirable. We addressed this question along the lines suggested by reviewer #3, who asks whether the “ATPase-inactive” complexes that we discarded in Fig. 4 might still be active in transport. For this, we analysed the fluorescence traces of all the complexes that did not show the initial ATP induced quenching of BtuF-AF488. Our hypothesis was that these complexes are not active in ATP hydrolysis, and should be unable to transport B12. Indeed, we did not observe dequenching of BtuF-AF488 (i.e. rise in fluorescence) in any of the complexes not showing an initial decrease (Supplementary Fig. 11). Although we do not want to over interpret this experiment, we now discuss it explicitly and also emphasize that our interpretation is formally not entirely conclusive – but why it is the most parsimonious one.

It is unknown whether previous ensemble studies were probing the same (active) fraction, as such studies reveal average values over the entire population and cannot observe heterogeneity. However, the fact that we observed heterogeneity in multiple assays (ATPase with/without substrate, vitamin transport), and also other techniques, such as atomic force or cryo-electron microscopy, have revealed numerous examples of heterogeneity, it is likely that heterogeneity was also present in all ensemble experiments, albeit invisible.

Revised text:

Page 5:

“Fluorescence traces of other (presumably inactive) complexes do not show any ATP-dependent features that can be discriminated from control experiments, and therefore we consider only the quenching as indicator for ATPase activity.”

Page 11:

“Independent of substrate concentration, fluorescence traces of complexes that do not show an initial ATP-dependent quenching effect remain flat, indicating that those transporters are unable to translocate vitamin (Supplementary Fig. 11).”

Page 13:

“Formally, it cannot be excluded that in Fig. 4 we probe a fraction of transporters – different from the fraction with high basal ATPase-activity fraction (Fig. 1) – that exhibits strong coupling between ATP turnover and transport. Since ATP turnover of this hypothetical coupled fraction would be at least hundred-fold slower than the fraction with high ATP hydrolysis activity, this fraction would be invisible in Fig. 1. However, this assumption would mean that complexes showing ATP induced transmembrane conformational changes are transport-inactive. Moreover, the observed response times of ATPase activity (Fig. 1) and ATP induces quenching of Alexa Fluor 488-labeled BtuF (Supplementary Fig. 11) are similar. Therefore, the most parsimonious interpretation is that the highly ATPase-active and transport-active fractions overlap.”

Along the same line, the title is still misleading to the general readership as no conclusions on single translocation events or a productive transport cycle can be made. Based on fairly ill-defined reconstitution, lipid environment or transport mechanism, BtuCD-F requires 30 or even much more ATP hydrolysis to occasionally move one vitamin B12 molecule across the membrane, depending with lab performed the assays. Therefore, the study does not report on the productive transport mechanism but rather on futile cycles of BtuCD-F. The title should read 'single-molecule visualization of ATP hydrolysis events within the vitamin B12 ABC importer BtuCD-F'.

We agree that at this moment we cannot directly measure the coupling between ATP hydrolysis and transport, but we can detect both processes. We therefore updated the title to:

"Single-molecule visualization of ATP hydrolysis and substrate transport in the vitamin B12 ABC importer BtuCD-F"

Reviewer #2 (Remarks to the Author):

I thank the authors for their useful responses and revisions in the text and presentation of data. I have two important remaining questions, which are included below.

1. "Under conditions in which active transport takes place, a vitamin B12 molecule remains bound to the protein complex for tens of seconds, during which several ATP hydrolysis cycles can take place, before it is being transported across the membrane." As the ATPase measurements are done with a population of transporters of which 18% is transport-active (line 118), how do the authors know that vitamin B12 binding and multiple rounds of ATPase are directly linked and occur at the same transport complex? In other words, could the reconstituted proteins that are probed in ATPase measurements contain different sub-populations, and could non-transport-active complexes still exhibit ATPase activity? Given the published data on ABC transporters, for which even isolated NBDs exhibit ATPase activity, this would not be an unreasonable assumption.

See response to reviewer #1

Along similar lines: how do the authors know in line 186 "15 ATP molecules [hydrolyzed] per BtuCD-F complex" that it is the full complex that gives rise to ATP hydrolysis in the experiments? As this could be perceived as a weak point in the paper, it will be important for the authors to discuss this further in the main text.

As shown by Fig. 1 and 2, BtuCD-F forms a stable complex under conditions that we used throughout our study (3 μ M BtuF). We therefore assume that it is the full complex that gives rise to this ATPase turnover rate. We added to the legend of Fig. 1:

"(b) ... When BtuF is present at the concentrations used, the full complex is formed."

2. Could the suggestion of uncoupled transport (line 282-283) also be explained if BtuCD-F would transport alternative substrates in the test system, in addition to vitamin B12? The authors refer to P-glycoprotein (line 320-321) for which transport of lipids and detergents in the experimental system has been raised as an explanation for the basal ATPase activity in the absence of drugs. Is this type of explanation

relevant for your research data? This point requires discussion in the manuscript.

Currently, only two substrates are known to be transported by BtuCD-F: cobolamin and the chemically related cobinamide (Mireku *et al.*, 2017), and therefore we think the above-mentioned explanation is unlikely. We added to the discussion:

“Although the basal ATPase activity in the P-glycoprotein might be explained by the large variety of substrates it can transport, only two substrates are known to be imported by BtuCD-F: cobalamin (vitamin B12) and the chemically related cobinamide [29]. Therefore, the basal ATPase activity must be explained by other mechanisms.”

Reviewer #3 (Remarks to the Author):

The experimental approaches in this manuscript represent interesting and potentially fruitful lines of investigation that enable the functions of the BtuCD-F system to be quantified at the single-molecule scale. Yet, the nature of the signals obtained, which are based principally on fluorescence intensity changes, severely limit the analyses presented and largely reduce the conclusions to qualitative correlations. While ATP-turnover assays have been added, and the authors have made a significant efforts to address the majority of the concerns put forward during initial review, the main conclusions of the manuscript seem to indicate what has been put forward previously: that BtuCD futilely hydrolyzes ATP in the absence of substrate and that BtuF binds stably to BtuCD (the idea that it is a breakthrough that this occurs in a lipid bilayer and not just detergent micelles seems questionable). Although the authors have commented on the observation (and critique) that the BtuCD/F system burns ATP non-productively to suggest that this may be advantageous to vitamin scavenging. Such a model would appear to be an outlier with respect to what has been shown for other biological systems, where energy conservation and energy efficiency are put forward as paramount to evolutionary fitness. Supporting this view, small molecules that uncouple energy expenditure from function in living systems are typically antibiotics/antiproliferative agents.

Even though we are unable to show direct measurements of coupling between ATPase activity and substrate transport, work from several different group has shown that this transporter turns over ATP in a “futile” way. This could be considered an outlier to other biological systems, however, it must be put in perspective. For example, considering the protein synthesis machinery in an *E. coli* cell: thousands of ribosomes synthesise 10 amino acids per second each, with 4 high energy phosphate bonds required per peptide bond. This easily adds up to $\sim 10^6$ ATP equivalents per second, which is several orders of magnitude higher than the energy consumption by BtuCD-F, making it doubtful whether optimizing energy efficiency of this and a few other transporters confers evolutionary advantages. In the discussion we now state:

“Possibly there is also no evolutionary drive to optimize transporters for rare substrates as total ATP consumption for these transporters is negligible compared to total cellular ATP consumption.”

The other key conclusion presented, which is based in part on what the authors refer to as a three color experiment (Fig.3) is that BtuF stays bound to BtuCD while ATP hydrolysis occurs. This conclusion is inferred from their measurements indicating that ATP turnover occurs relatively quickly (on the order of seconds time scale fluorescence fluctuations) and the multicolor imaging studies discussed surrounding Fig.3, which have been interpreted as providing the conclusive evidence that BtuF dissociation is slow (on the order of minutes). As it stands Fig.3 is unconvincing,

however: there are only two traces shown, neither of which shows a photobleaching event to confirm that FRET was indeed being observed; and if the conclusion is that 488-555 traces and 555-647 traces show highly stable complexes and no evidence of exchange, why not show one of each kind of trace to substantiate claim that this is a three-color experiment?

With respect to the three-color FRET experiments, we are unfortunately technically limited to only visualize acceptor emission of 555 and 647 (using either 488, 555 or 647 excitation). The traces presented show simultaneous emission of both acceptors on BtuF via the 488 (donor) on BtuC excitation. The fact that traces with either 555 or 647 emission are found upon 488 excitation, indicates we are observing FRET.

Although we cannot observe 488 emission in these experiments directly (for technical reasons, inherent to the microscopy set-up), we performed experiments to confirm the distribution of fluorophores and transporters in liposomes (data not shown, but similar to Supplementary Fig. 3). As we are looking at binary FRET, this does not complicate our data. Below, two panels are shown with 555 bleaching (left, around 150 seconds) and 647 bleaching (right, around -30 seconds). Bleaching traces are sparse, as we use low laser power for sustained fluorescence.

We modified parts of the results section (page 9).

An alternative, bulk method like liposome pelleting or pull down would seem necessary to provide supporting evidence for the single-molecule interpretation that the complex is highly stable.

We have considered these type of experiments, but results of such experiments would be unreliable, they inevitably require a step in which the effective lower to outside concentration would drop (far) below the K_d . Bulk methods have already been performed by Korkhov *et al.* (2014), but were not interpreted as such. The main text states:

“Our findings are strongly supported by previous ensemble characterizations of the transporter embedded in liposomes where the authors looked at exchange of labelled versus unlabelled BtuF and no exchange was observed [16].”

While there is little doubt that the efforts presented are substantial and unique, these questions about the nature and novelty of the conclusions drawn brings up one of the foundational concerns I have about this work: why is only 8% (or 18% now that sidedness has been taken into consideration) of the protein included in the authors’ analysis. While it is appreciated that others have reported similar difficulties working with this system, have previous structural and functional analyses of BtuCD(F) (i.e.

like the ATPase assay in Figure 1) only focused on such a small subpopulation of active transporters?

While the authors focus on this subpopulation because it is the only fraction that provides the futile ATP hydrolysis signal, and I cannot argue that the conclusions drawn from this subpopulation are what they are, I am left with the sense that the present body of work would benefit from functional analyses of majority of molecules that are currently discarded. For instance, how do the single-molecule ATP hydrolysis assays compare to the bulk studies: does the majority of protein hydrolyze ATP or only the small subpopulation? Do the majority of proteins transport (ie. do the authors sees the rise if fluorescence show in Fig.4b for all proteins or for just an 8% subpopulation)? Is it possible that the conclusions that would be drawn about the majority of protein would be different than the 8%, for instance, does this fraction have a tighter coupling between ATP binding/hydrolysis and transport and ATP hydrolysis events are just not observed because the signal-averaging procedures employed mask the single (or few) fluorescence fluctuations that are present?

See response to reviewer #1

In summary, although the revised manuscript is indeed substantially improved by the new experiments and discussion that have been included, in my view the aforementioned issues, together with the general considerations that there is little to effort made to show the reproducibility or statistical significance of the findings presented and profoundly new and obvious biological insights are lacking, stipulate that further refinements to the manuscript are needed prior to publication.

REVIEWERS' COMMENTS:

Reviewer #1 (Remarks to the Author):

As previously stated, the authors strove to cover most of the concerns raised by the reviewers in their revised manuscript. However, one major – and essential – area of criticism remains unaddressed.

The work reports mainly on conformational changes detected by fluorescence quenching, which are interpreted as single ATP hydrolysis events. A convincing correlation of these events at different ATP concentrations could not be proven. Based on the indirect observation of single turnover events, the correlation with the ATPase activity at various ATP concentrations is essential. Unfortunately, the controls using non-hydrolysable ATP or an ATPase inactive mutant are not sufficient to report on the important dose-dependent and potential rate-limiting step of the ATPase cycle. Varying the ATP concentration offers an elegant way to change the time regime of the fluctuations.

The newly proposed title "Single-molecule visualization of ATP hydrolysis and substrate transport in the vitamin B12 ABC importer BtuCD-F" is not supported by the experimental data and therefore largely misleading as it implies that both processes are coupled or were analyzed at the same time. No direct conclusion on productive single translocation events can be drawn. The authors already admitted that the correlation between fluorescence fluctuations and B12 transport is rather indirect and formally not entirely conclusive.

An appropriate title would be: "Single-molecule visualization of conformational changes in the vitamin B12 ABC importer BtuCD-F".

Reviewer #2 (Remarks to the Author):

I thank the authors for their efforts to address the questions raised by myself and the other reviewers. It is technically really difficult to address the question whether the ATPase activity measured, originates from the full transport-active complex only. The authors should perhaps add a sentence to the discussion to acknowledge that this point could affect conclusions regarding the ratio of the number of ATP molecules hydrolysed per transported substrate. However, given the technical advance of this work overall, it is my opinion that the authors should have the opportunity to publish this manuscript with this adjustment.

Reviewer #1 (Remarks to the Author):

As previously stated, the authors strove to cover most of the concerns raised by the reviewers in their revised manuscript. However, one major – and essential - area of criticism remains unaddressed.

The work reports mainly on conformational changes detected by fluorescence quenching, which are interpreted as single ATP hydrolysis events. A convincing correlation of these events at different ATP concentrations could not be proven. Based on the indirect observation of single turnover events, the correlation with the ATPase activity at various ATP concentrations is essential. Unfortunately, the controls using non-hydrolysable ATP or an ATPase inactive mutant are not sufficient to report on the important dose-dependent and potential rate-limiting step of the ATPase cycle. Varying the ATP concentration offers an elegant way to change the time regime of the fluctuations.

We agree that non-hydrolysable ATP and the ATPase inactive mutant are not sufficient to report on a dose-dependent ATPase cycle, but that is not what we aim for. We use these controls to prove that our observable is related to ATPase activity. Insight into dose-dependent behaviour could be informative, but is not necessary for the conclusions in our work. Given the challenges in performing these experiments, we believe that they are beyond the scope of the current study.

We added a short statement about this in the discussion:

“Is the ATPase activity only originating from transport-active complexes? A dose-dependent relation between ATPase activity and substrate transport could give insight.”

The newly proposed title “Single-molecule visualization of ATP hydrolysis and substrate transport in the vitamin B12 ABC importer BtuCD-F” is not supported by the experimental data and therefore largely misleading as it implies that both processes are coupled or were analyzed at the same time. No direct conclusion on productive single translocation events can be drawn. The authors already admitted that the correlation between fluorescence fluctuations and B12 transport is rather indirect and formally not entirely conclusive.

An appropriate title would be: “Single-molecule visualization of conformational changes in the vitamin B12 ABC importer BtuCD-F”.

Although in our opinion the conjunction “and” does not imply coupling or synchronization, we would like to avoid any possible misinterpretation and suggest the following title:

“Single-molecule visualization of conformational changes and substrate transport in the vitamin B₁₂ ABC importer BtuCD-F”

The visualization of substrate transport is a key observation in our work, and therefore must be included in the title, but by changing “ATP hydrolysis” to “conformational changes” the threat that readers may misinterpret the title should be eliminated.

Reviewer #2 (Remarks to the Author):

I thank the authors for their efforts to address the questions raised by myself and the other reviewers. It is technically really difficult to address the question whether the ATPase activity measured, originates from the full transport-active complex only. The

authors should perhaps add a sentence to the discussion to acknowledge that this point could affect conclusions regarding the ratio of the number of ATP molecules hydrolysed per transported substrate. However, given the technical advance of this work overall, it is my opinion that the authors should have the opportunity to publish this manuscript with this adjustment.

We have added a sentence to the discussion:

“..., but alternative interpretations could affect conclusions regarding the ratio of number of ATP molecules hydrolysed per transported substrate